# Mitigating Shortcut Learning with Diffusion Counterfactuals and Diverse Ensembles

## Abstract

Spurious correlations in the data, where multiple cues are predictive of the target labels, often lead to a phenomenon known as shortcut learning, where a model relies on erroneous, easy-to-learn cues while ignoring reliable ones. In this work, we propose *DiffDiv* an ensemble diversification framework exploiting Diffusion Probabilistic Models (DPMs) to mitigate this form of bias. We show that at particular training intervals, DPMs can generate images with novel feature combinations, even when trained on samples displaying correlated input features. We leverage this crucial property to generate synthetic counterfactuals to increase model diversity via ensemble disagreement. We show that DPM-guided diversification is sufficient to remove dependence on shortcut cues, without a need for additional supervised signals. We further empirically quantify its efficacy on several diversification objectives, and finally show improved generalization and diversification on par with prior work that relies on auxiliary data collection.

## 1 Introduction

Deep Neural Networks (DNNs) have achieved unparalleled success in countless tasks across diverse domains. However, they are not devoid of pitfalls. One such downside manifests in the form of shortcut learning, a phenomenon whereby models latch onto simple, non-essential cues that are spuriously correlated with target labels in the training data (Geirhos et al., 2020; Shah et al., 2020; Scimeca et al., 2022). Often engendered by the under-specification present in data, this simplicity bias presents easy to learn *shortcuts* that allow accurate prediction at training time, irrespective of a model's alignment to the downstream task. For instance, previous work has found models to incorrectly rely on background signals for object recognition (Xiao et al., 2021; Beery et al., 2018), or to rely on non-clinically relevant metal tokens to predict patient conditions from X-Ray images (Zech et al., 2018). Shortcut learning has often been found to lead to significant drops in generalization performance (Agrawal et al., 2018; Torralba & Efros, 2011; Minderer et al., 2020; Li & Vasconcelos, 2019; Zech et al., 2018). Leveraging shortcut cues can also be harmful when deploying models in sensitive settings. For example, shortcuts can lead to the reinforcement of harmful biases when they endorse the use of sensitive attributes such as gender or skin color (Wang et al., 2019; Xu et al., 2020; Scimeca et al., 2022).

Addressing simplicity biases in machine learning has been a focus of extensive research. Numerous studies have aimed to encourage models to use a broader and more diversified set of predictive cues, especially when dealing with training data that lacks explicit shortcut cue labels. A variety of these methods have been input-centric, designed to drive models to focus on different areas of the input space (Teney et al., 2022b; Nicolicioiu et al., 2023), while others have focused on diversification strategies that rely on auxiliary data for *prediction disagreement* (Pagliardini et al., 2022; Lee et al., 2022). The latter approaches, in particular, have been instrumental in developing functionally diverse models that exhibit robustness to shortcut biases. However, they are limited by their required access to auxiliary data that is often challenging to obtain.

The primary objective of this work is to mitigate shortcut learning tendencies, particularly when they result in strong, unwarranted biases, access to ood data is expensive, and different features may rely on similar areas of the input space. To achieve this objective, we propose *DiffDiv*, an ensemble framework relying on unlabelled *ood* data for shortcut mitigation by ensemble diversification. To overcome the challenges of the past, we aim to synthetically generate the data for model diversification

Figure 1: DiffDiv: We sample from a DPM to generate synthetic counterfactuals showcasing emergent novel feature combinations. These samples are then utilized to build a diverse model ensemble via different ensemble disagreement objectives.

by disagreement, and thus avoid the impracticality of *ood* data collection. We posit that the synthetic data should: first, lie in the manifold of the data of interest; and second, be at least partially free of the same shortcuts as the original training data. We leverage Diffusion Probabilistic Models (DPMs) to generate synthetic data for ensemble disagreement.

Although the in-depth study of the generalization properties of diffusion sampling mechanisms is beyond the scope of this paper, we make the crucial observation and empirically show that even in the presence of correlated features in the data, appropriately trained DPMs can be used to generate synthetic counterfactuals that break the shortcut signals present at training time. We show that this important characteristic arises at specific training intervals, and that it can be leveraged for shortcut cue mitigation. We hypothesize that diversification, and shortcut mitigation, can be achieved via ensemble disagreement on these DPM-generated samples, providing models with an opportunity to break the spurious correlations present during training. Remarkably, our experiments confirm that the extent and quality of our diffusion-guided ensemble diversification is on par with existing methods that rely on additional data.

Our contributions are the following:

1. We show that DPMs can generate feature compositions beyond data exhibiting correlated input features.

2. We demonstrate that ensemble disagreement is sufficient for shortcut cue mitigation.

3. We propose *DiffDiv*, a framework to achieve bias mitigation through diversification based on diffusion counterfactuals.

4. We show appropriately trained DPM counterfactuals can lead to state-of-the-art diversification and shortcut bias mitigation.

Moreover, our study presents several interesting findings, including the application of the Wisconsin Card Sorting Test for Machine Learners (WCST-ML) (Scimeca et al., 2022) to the CelebA face dataset for the first time, exposing biased inference for *pale skin* features.

## 2  RELATED WORK

### 2.1  OVERCOMING THE SIMPLICITY BIAS

**Deliberate de-bias:**    To overcome the simplicity bias, copious literature has explored methodologies to avoid or mitigate shortcut cue learning when labels for the shortcut cues were present in the training data (Li & Vasconcelos, 2019; Kirichenko et al., 2023; Wang et al., 2019; Kim et al., 2019; Sagawa* et al., 2020; Lee et al., 2021). The access to shortcut signals is, however, a critical limitation, as these are generally hard or impossible to obtain.

**Data augmentation:**    Data augmentation methodologies have proven useful in the generation of *bias-conflicting* or bias-free samples (Kim et al., 2021; Lim et al., 2023; Jung et al., 2023), or to augment underrepresented subgroups therein (Wang et al., 2020; Mondal et al., 2023), leading to less biased predictions. Although an important research direction, assumptions on the nature of the

biases are here still necessary, and the rejection of selected cues for prediction may not always lead to improved generalization, notably when the downstream task is aligned with said cues.

**Diversification:** A different approach to this problem has been to enforce the use of a diverse set of signals for prediction. The use of ensembles has been one such method, where models' diversity would lead to bias mitigation and improved generalization. Different weight initialization and architectures have previously been shown to be ineffective in the presence of strong shortcut biases (Scimeca et al., 2022). Methods ensuring mutual orthogonality of the input gradients have proven more effective, driving models to attend to different locations of the input space for prediction (Ross et al., 2020; Teney et al., 2022a;b; Nicolicioiu et al., 2023). These input-centric methods, however, may be at a disadvantage in cases where different features must attend to the same area of the input space. Instead, a different approach has been via the diversification of direct ensemble model predictions. This approach hinges on the availability of –unlabelled– out-of-distribution (*ood*) auxiliary samples that are, at least in part, free of the same shortcuts as the original training data. Through a diversification objective, the models are then made to disagree on these *ood* samples, while maintaining performance on the original data, effectively fitting functions with different extrapolation behaviors (Lee et al., 2022; Pagliardini et al., 2022; Scimeca et al., 2023; Lin et al., 2023). Even in this case, the auxiliary *ood* data dependency poses limitations, as this is often not readily accessible, and can be costly to procure.

## 2.2 DPM Modelling and Generalization

In recent years, Diffusion Probabilistic Models have emerged as a transformative generative tool, spearheading progress in the generative domain across various applications, including vision (Ho et al., 2020; Song & Ermon, 2019), efficient sampling methods (Sendera et al., 2024), and text (Gong et al., 2022; Venkatraman et al., 2024) and even control tasks (Venkatraman et al., 2024). Numerous studies have underscored their prowess in generating synthetic images, which can then be harnessed to enrich datasets and bolster classification performance (Sariyildiz et al., 2023; Azizi et al., 2023; Yuan et al., 2022; Dunlap et al., 2023; Howard et al., 2023).

In several cases, DPMs have been shown to transcend the surface-level statistics of the data, making them invaluable in understanding data distributions and features (Chen et al., 2023; Yuan et al., 2022; Wu et al., 2023). Moreover, recent studies have indicated the ability of DPMs to achieve feature disentanglement via denoising reconstructions (Kwon et al., 2022; Wu et al., 2023; Okawa et al., 2023). In particular, work in (Wang et al., 2023) has shown how text-guided generative models can represent disentangled concepts, and how, through algebraic manipulation of their latent representations, it is possible to compositionally generalize to novel and unlikely combination of image features.

Additional work has also shown strong inductive Diffusion biases during learning, which may explain some of these phenomena (Kadkhodaie et al., 2023). In our work, we test the edge case of DPMs trained with data exhibiting correlated input features. We find that when suitably trained, DPMs can still generate samples with novel feature combinations and that these can be leveraged for ensemble diversity.

## 3 Methods

### 3.1 DiffDiv Overview

We apply DiffDiv in two stages. The first stage corresponds to training a DPM on the dataset of interest. We will explain in later sections how this training can often be interrupted early, to allow for the DPMs to generate samples especially useful for diversification. The second stage corresponds to training an ensemble on the dataset and task of interest. This training entails the joint optimization of two objectives, a standard classification objective, and a diversification objective. The standard training objective is performed on real data, while the diversification objective is performed on synthetic counterfactuals generated by the pre-trained DPM. We only consider a fixed small set of 3k counterfactuals in all experiments, limiting the necessity for expensive DPM generation to apply our methods. We refer the reader to the supplementary results for ablations on counterfactual set size.

The following sections describe the key components in DiffDiv. In §3.2 we briefly summarize diffusion training and sampling during phase 1. In §3.3 we present the diversification objectives considered for ensemble training in phase 2. In §3.4, as a testbed for our experiments, we introduce an extreme data setup where the task labels are fully correlated with each of the input features. And finally, in §3.5, we introduce the datasets considered in our experiments.

## 3.2 DPMs and Efficient Sampling

We utilize Diffusion Probabilistic Models (DPMs) to generate synthetic data for our experiments. DPMs operate by iteratively adding or removing noise from an initial data point $x$ through a stochastic process governed by a predefined noise schedule. Let $\mathbf{z} = \{\mathbf{z}_t \mid t \in [0, 1]\}$ be a latent variable conditioned on $t$, and characterized by a noise-to-signal ratio $\lambda_t = \log\left[\alpha_t^2/\sigma_t^2\right]$ decreasing monotonically with $t$. In the forward process, noise is added to $x$ to transform it into $z_t$ (Salimans & Ho, 2022):

$$q\left(\mathbf{z}_t \mid \mathbf{x}\right) = \mathcal{N}\left(\mathbf{z}_t; \alpha_t \mathbf{x}, \sigma_t^2 \mathbf{I}\right), \quad q\left(\mathbf{z}_t \mid \mathbf{z}_s\right) = \mathcal{N}\left(\mathbf{z}_t; (\alpha_t/\alpha_s)\mathbf{z}_s, \sigma_{t|s}^2 \mathbf{I}\right) \tag{1}$$

where and $0 \leq s < t \leq 1$, and $\sigma_{t|s}^2 = \left(1 - e^{\lambda_t - \lambda_s}\right)\sigma_t^2$. We let $\alpha_t$ follow a cosine schedule, thus $\alpha_t = \cos(0.5\pi t)$.

The reverse process then aims to reconstruct $x = z_0$ by iteratively denoising $z_t$ into $z_s$, starting from $z_1 \sim \mathcal{N}(0, I)$. To facilitate efficient sampling, we employ Denoising Diffusion Implicit Models (DDIM) (Song et al., 2020), a first-order ODE solver for DPMs (Salimans & Ho, 2022; Lu et al., 2022), utilizing a predictor-corrector scheme to minimize the number of sampling steps:

$$\begin{aligned}
\mathbf{z}_s &= \alpha_s \hat{\mathbf{x}}_\theta\left(\mathbf{z}_t\right) + \sigma_s \frac{\mathbf{z}_t - \alpha_t \hat{\mathbf{x}}_\theta\left(\mathbf{z}_t\right)}{\sigma_t} \\
&= e^{(\lambda_t - \lambda_s)/2}\left(\alpha_s/\alpha_t\right)\mathbf{z}_t + \left(1 - e^{(\lambda_t - \lambda_s)/2}\right)\alpha_s \hat{\mathbf{x}}_\theta\left(\mathbf{z}_t\right)
\end{aligned} \tag{2}$$

The term $\hat{\mathbf{x}}_\theta\left(\mathbf{z}_t\right)$ is the neural network's output, predicting the denoised data from the noisy observation at timestep $t$. And train the denoising model by:

$$L_\theta = \left(1 + \frac{\alpha_t^2}{\sigma^2}\right)\|\mathbf{x} - \hat{\mathbf{x}}_t\|_2^2 \tag{3}$$

In our setup, we consider DPMs trained at different fidelity levels, postulating that, generally, increased DPM training will more closely fit the data distribution at hand. Therefore, in our experiments, we use the 'number of diffusion training epochs' as a proxy for the DPM fidelity on the modeled distribution.

## 3.3 Ensemble Training and Diversification

In our setup, we wish to train a set of models within an ensemble while encouraging model diversity. Let $f_i$ denote the $i^{th}$ model predictions within an ensemble consisting of $N_m$ models. Each model is trained on a joint objective comprising the conventional cross-entropy loss with the target labels, complemented by a diversification term computed on synthetic counterfactuals, represented as $L_{\text{div}}$. The composite training objective is:

$$\mathcal{L} = \mathcal{L}_{\text{xent}} + \gamma \, \mathcal{L}_{\text{div}}^{\text{obj}} \tag{4}$$

where $\mathcal{L}_{\text{xent}}$ is the cross-entropy loss, $\mathcal{L}_{\text{div}}^{\text{obj}}$ is the diversification term for a particular objective, and $\gamma$ is a hyper-parameter used to modulate the importance of diversification within the optimization objective. To impart diversity to the ensemble, we investigated five diversification objectives denoted by $obj \in \{$ div, cross, $L_1$, $L_2$, kl$\}$.

The $L_1$ and $L_2$ baseline objectives were designed to induce diversity by maximizing the distance between each model output and the moving average of the ensemble prediction, thus:

$$L_{\text{reg}}^{L_1} = -\frac{1}{N_m} \sum_{i=1}^{N_m} \left\| f_i - \frac{1}{N_m} \sum_{j=1}^{N_m} f_j \right\|_1 \quad L_{\text{reg}}^{L_2} = -\frac{1}{N_m} \sum_{i=1}^{N_m} \left\| f_i - \frac{1}{N_m} \sum_{j=1}^{N_m} f_j \right\|_2^2$$

The *cross* objective diversifies the predictions by minimizing the negative mutual cross-entropy of any two models:

$$\mathcal{L}_{\text{reg}}^{\text{cross}} = -\frac{1}{N_m(N_m - 1)} \sum_{i \neq j} \frac{\text{CE}(f_i, \text{argmax}(f_j)) + \text{CE}(f_j, \text{argmax}(f_i))}{2}$$

Finally, the *kl* objective aims at maximizing the *kl* divergence between the output distributions of any two models, while the *div* diversification objective is adapted from (Lee et al., 2022) and encourages diversity by minimizing the mutual information of any two models' predictions; they can be summarized as:

$$\mathcal{L}_{\text{reg}}^{kl} = -\frac{1}{N_m(N_m - 1)} \sum_{i \neq j} D_{KL}(f_i \| f_j) \quad \mathcal{L}_{\text{reg}}^{\text{div}} = \frac{1}{N_m(N_m - 1)} \sum_{i \neq j} D_{KL}(p(f_i, f_j) \| p(f_i))$$

### 3.4 WISCONSIN CARD SORTING TEST FOR MACHINE LEARNERS (WCST-ML)

To isolate and investigate shortcut biases, we employ the Wisconsin Card Sorting Test for Machine Learners (WCST-ML), a method devised to dissect the shortcut learning behaviors of deep neural networks (Scimeca et al., 2022). We use the splits from WCST-ML to both train and evaluate the ensembles in this work, as it provides a systematic approach to creating datasets with multiple cues, designed to correlate with the target labels. Specifically, given $K$ cues $i_1, i_2, \ldots, i_K$, the method produces a *diagonal* dataset $\mathcal{D}_{\text{diag}}$ where each cue $i_k$ is equally useful for predicting the labels $Y$, with the total number of classes $L = |Y|$. This level playing field is instrumental in removing the influence of feature dominance and spurious correlations, thereby allowing us to observe a model's preference for certain cues under controlled conditions. To rigorously test these preferences, WCST-ML employs the notion of *off-diagonal* samples. These are samples where the cues are not in a one-to-one correspondence with the labels, but instead align with only one of the features under inspection. By evaluating a model's performance on off-diagonal samples, according to each feature, we can test and achieve an estimate of a model's reliance on the same.

### 3.5 DATASETS

As the experimental grounds for our study, we leverage three representative datasets: a color-augmented version of DSprites (Matthey et al., 2017), UTKFace (Zhang et al., 2017), and CelebA (Liu et al., 2015). The choice of the datasets in our work was due to several factors; most prominently, the *disentangled* nature of the features in DSPrites – providing a particularly useful analysis with a controlled overlap of bias tendencies –, UTKFace, a dataset previously known to present strong ethnical bias in WCST-ML, with concerning societal implications (Scimeca et al., 2022), and CelebA, providing a large scale more complex and realistic setting to benchmark DiffDiv.

OPERATIONALIZING WCST-ML ACROSS DATASETS

We train our models on the correlated WCST-ML diagonal sample sets for each dataset. For ColorDSprites, we consider $K_{DS} = 4$, features $\{color, orientation, scale, shape\}$, and $L = 3$ as constrained by the number of shapes in the dataset. Within UTKFace we consider $K_{UTK} = 3$, features $\{ethnicity, gender, age\}$, and $L = 2$ as constrained by the binary classification on *gender*. In CelebA we consider $K_{UTK} = 2$, features $\{lightskin, ovalface\}$, and $L = 2$ due to the binary classification on all features. See §S1 for additional implementation details.

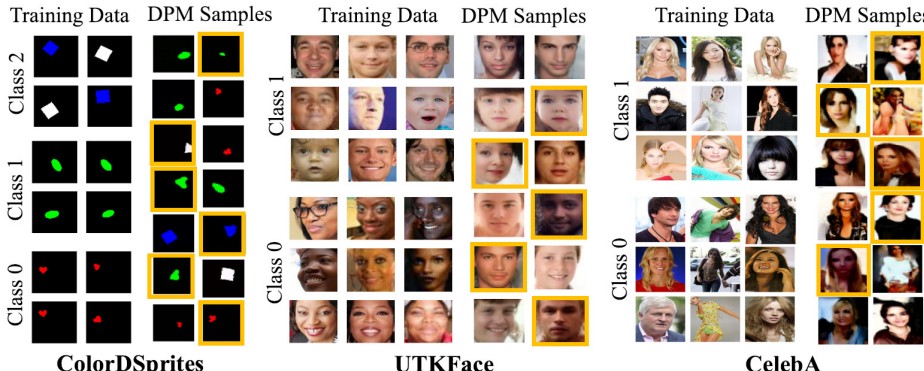

Figure 2: DPM training and counterfactual generation. While training on images showcasing a correlated set of features (left columns), sampling from DPMs at appropriate fidelity levels can generate novel objects beyond the combinations of features observed during training (marked right-hand side images).

# 4 RESULTS

## 4.1 IMPLEMENTATION

To investigate the objectives of this study, we apply the WCST-ML framework on ColorDSprites, UTKFace and CelebA, creating training datasets of fully correlated feature-labels groups (Fig. 1). For ColorDSprites, we consider the features of $K_{DS} = \{color, orientation, scale, shape\}$, in UTKFace we consider the features $K_{UTK} = \{ethnicity, gender, age\}$, while in CelebA we consider the features $K_{UTK} = \{lightskin, ovalface\}$.

We train three DPMs on the *diagonal* fully-correlated sets for each dataset, including respectively 34998, 1634 and 2914 feature-correlated samples. As mentioned in §4.1, we consider the number of DPM training epochs to be a proxy of the diffusion model's fidelity, or closeness, to the distribution of interest, where longer training generally leads to higher generative consistency of the target distribution. We generate $\approx$ 100k samples from DPMs trained at varying number of epochs between 1 to 1.2K, to be used for ensemble diversification and analysis. We perform no post-processing or pruning of the generated samples, and instead wish to observe the innate ability of DPMs to generate samples beyond the training distribution.

For each of the ensemble experiments, we train a diverse ensemble comprising 100 ResNet-18 models on all datasets. We perform ensemble training separately on all the considered objectives, and for each, we perform ablation studies by applying the diversification objective to the data generated by the DPMs at different fidelity levels. For comparison, we also consider ensemble diversification with real *ood* data, randomly sampled from the *off-diagonal* sets, as well as a standard ensemble baseline with no diversification.

## 4.2 DIFFUSION COUNTERFACTUAL SAMPLING

**DPMs Exhibit Generalization Capabilities Under Correlation:** We tested the ability of DPMs to transcend surface-level statistics of the data and generate samples that break the shortcut signals at training time, even when trained on correlated data. Fig. 2 displays the training samples for both datasets (left halves), as well as samples from the DPMs (right halves) trained for 25 and 800 epochs, respectively. In the figure, we observe how sampling from the trained DPMs generates previously unseen feature combinations, despite the correlated coupling of features during training (e.g. $\langle green/white, heart \rangle$, $\langle child, female \rangle$ or $\langle \neg light\ skin, oval\ face \rangle$), suggesting a potential for DPMS to transcend surface-level statistics of the data, confirming and further extending previous findings to the special case of fully correlated input features in the target distribution. We leverage this important characteristic in the generation of samples for disagreement, as it allows models to break from the shortcuts in the training distribution. Importantly, as later shown, we consider disentangled samples from non-fully converged DPMs, trading sample fidelity for sample novelty, to appropriately induce diversification.

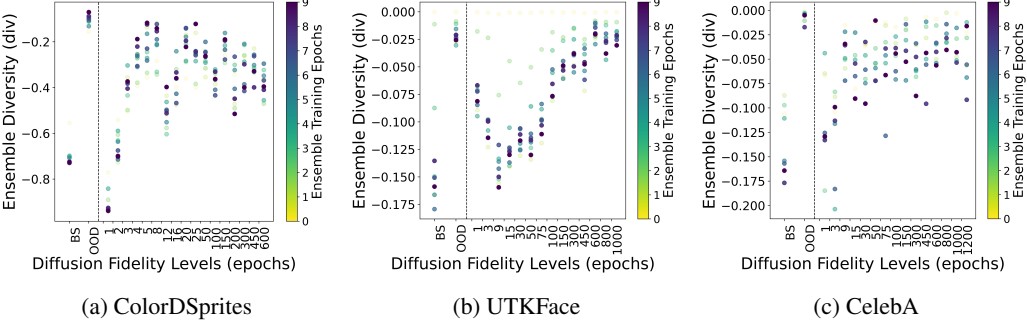

|                |              |            |
| :---: | :---: | :---: |
| (a) ColorDSprites | (b) UTKFace | (c) CelebA |

Figure 4: Prediction diversity when diversifying via samples from diffusion models at different fidelities (training epochs), as compared to using real *ood* samples (ood), and a baseline without diversification (BS).

**Early Stopping to Capture Diffusion ood Sampling Capabilities:** To understand under which conditions sampling from DPMs can lead to the generation of samples displaying unseen feature combinations, we examine the fraction of *ood* samples generated by DPMs at different fidelities (§4.1). ColorDSprites dataset is particularly useful for this analysis, given the disentangled nature of its features. To measure the fraction of *ood* samples, we train a near-perfect oracle on the full ColorDSprites dataset, trained to predict the WCST-ML partitioned labels under all features. We can then consider in-distribution (*id*) those samples showcase fully correlated feature levels (close to the diagonal samples), and out-of-distribution (*ood*) those samples classified to belong to the off-diagonal WCST-ML set.

Fig. 3 shows the fraction of *ood* samples generated by the ColorDSprites DPM at varying levels of training fidelities. We identify at least three qualitative different intervals. An initial *burn-in* interval, characterized by a high frequency of *ood* generated samples, but which fails to capture the manifold of the data; an *originative* interval, where we observe a reduced number of *ood* samples, in favor for a generative distribution more aligned with the data to represent; and an *exact* interval, where the DPM's ability to almost perfectly represent the data comes at the cost of novel emergent feature mixtures. In the context of ensemble diversification and bias mitigation, the *originative* interval is posed to provide the necessary information to break the simplicity shortcuts, while providing effective disagreement signals with respect to the visual cues available. We refer the reader to Suppl. §S2 for further insights into DPM-early stopping with ensemble diversification criteria. Interestingly, we will later show that even low-fidelity samples can induce appropriate diversification within *DiffDiv*.

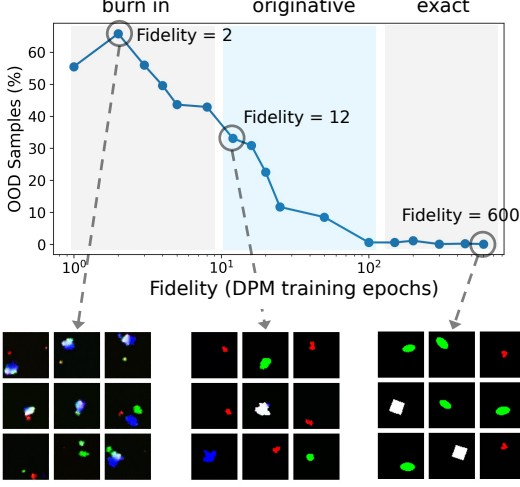

Figure 3: *ood* sample frequency for DPMs trained at different fidelities on ColodDSprites. Three intervals are prominent: *burn-in*, characterized by high *ood* sample frequency, but poor sample quality; *originative*, where the model has learned the manifold of the data while still displaying capabilities for *ood* sample generation; and *exact*, characterized by near-perfect samples, but a significantly reduced *ood* sample generation.

## 4.3 DIFFUSION-GUIDED ENSEMBLE DIVERSITY

We test whether diffusion counterfactuals can lead to ensemble diversification. To do so we trained ensembles comprising 100 ResNet-18 models on ColorDSprites, UTKFace, and CelebA. The ablation studies examined the training of the ensemble with each diversification objective in turn, as well as a baseline with no imposed diversification. For both baseline and diversification experiments, we favor different training dynamics by using separate vanilla Adam optimizers for each model. We train the ensemble with a cross-entropy loss on the correlated diagonal training data, as well as an

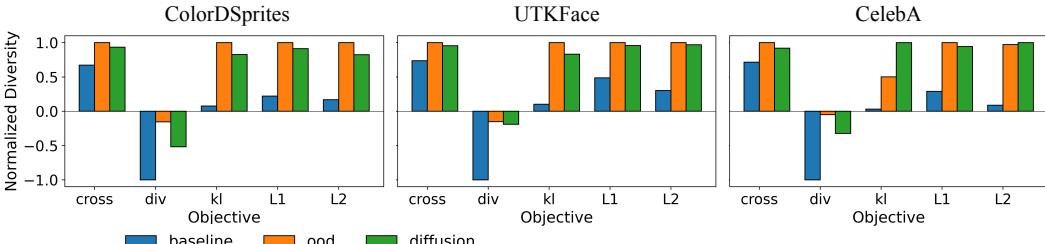

Figure 5: Diversity comparison for ensemble trained on *ood* data and diffusion counterfactuals across metrics (higher is better).

additional, objective-specific, diversification loss, computed on a separate diversification set. In our experiments, we will refer to '*ood*' to the case where the data for disagreement is comprised of left-out, feature-uncorrelated, samples from the original dataset; to 'diffusion' when the samples are generated by a DPM; and to 'baseline' when no diversification objective is included. We tune $\gamma$ by grid-search to provide comparable experiments under different objectives (see Suppl. §S2.1)

**DPM Fidelity Significantly Impacts Diversification:** We consider the case where the diversification objective is computed on samples generated by a DPM at different fidelity levels. We show in Fig. 4 the diversity achieved on the *div* diversification objective when training the ensemble on the same, with varying diffusion fidelity samples, and report the full set of diversification fidelity experiments in Suppl. Fig S3. In both Fig. 4a and Fig. 4b we observe the diversity of the ensemble predictions to vary significantly, from the low-diversity baseline (BS), trained without a diversification set, to the high diversity achieved with *ood* samples. For both datasets, extremely low fidelities provide little use for diversification, leading to low-diversity predictions by the ensembles. The DPM trained on ColorDSprites, however, quickly fits the synthetic data distribution and provides counterfactual samples achieving similar diversification performance to the original off-diagonal samples at early fidelity levels. Ultimately, excessive diffusion training leads to limited *ood* sample generation and lower diversification performance. The DPMs trained on UTKFace and CelebA train longer to appropriately model the more complex distributions, achieving comparable diversification levels to the *ood* sample set after approximately 800 and 1000 training epochs respectively. Importantly, we find that appropriate DPM training and early stopping procedures are necessary to generate samples that capture the distribution at hand, while still displaying novel sampling behaviors. We expand on early stopping signals for DPM training in Suppl. §S2.3.

**Diffusion-guided Diversity Leads to Comparably Diverse Ensembles:** Given the difficulty and cost of collecting auxiliary *ood* data for diversification, we wish to compare the level of diversification achieved with DPM counterfactual, as opposed to using real *ood* data. Based on our previous results in Fig. 4 (also Suppl. Fig. S3), we choose samples drawn from DPMs trained for 100, 800 and 1200 epochs for ColorDSprites, UTKFace and CelebA respectively, providing samples especially useful for diversification. In Fig. 5 we report the objective-wise normalized diversity achieved in each scenario by the ensemble. We find that diffusion counterfactuals can lead to comparable diversification performance with respect to real *ood* samples. In the figure, the diffusion-led diversity is almost always within 5% from the metrics achieved when using pure *ood* samples, both typically over 50% higher than the baseline.

**Ensemble Diversification Breaks Simplicity Biases:** By WCST-ML, we can test each model's bias to a cue by testing the model's output on purposefully designed test sets (Scimeca et al., 2022). We test the quality of the diversification obtained reporting in Table 2 the fraction of models attending to specific cues, when trained on real *ood* data and diffusion-generated samples (*DiffDiv*). A model was deemed to be attending to a cue if its validation accuracy on the cue-specific *ood* WCST-ML dataset was highest relative to all other features. Firstly, our baseline findings mirrored the observations in (Scimeca et al., 2022). Specifically, models trained on ColorDSprites under WCST-ML with no diversification (baseline) attend to *color*

Table 1: Average change in ensemble accuracy over the averted cues following diversification (reported in %).

| Obj. ↓ Dataset → | Color | Face | CelebA |
|---|---|---|---|
| Cross | 5.17 | 3.37 | 2.00 |
| Div | 8.16 | 3.61 | 0.45 |
| KL | 2.87 | 4.54 | 2.02 |
| L1 | 3.31 | 3.61 | 1.30 |
| L2 | 4.67 | 5.04 | 1.34 |

Table 2: Comparison between diversification with OOD samples and DiffDiv with model disagreement on five objectives. The feature columns report the fraction of models (not accuracy) attending to the respective features. The final column reports the average validation accuracy on the original training data. The shortcut feature for each dataset is highlighted in bold.

| Dataset → | | ColorDSprites | | | | | UTKFace | | | | CelebA | | |
|---|---|---|---|---|---|---|---|---|---|---|---|---|---|
| Algo ↓ Feat. | Obj. ↓ → | **Color** ($\downarrow$) | Orient. | Scale | Shape | Acc. ($\uparrow$) | Age | **Ethnicity** ($\downarrow$) | Gender | Acc. ($\uparrow$) | Oval Face | **Pale Skin** ($\downarrow$) | Acc. ($\uparrow$) |
| Baseline | - | 1.00 | 0.00 | 0.00 | 0.00 | $1.000_{\pm0.00}$ | 0.00 | 1.00 | 0.00 | $0.920_{\pm0.02}$ | 0.00 | 1.00 | $0.857_{\pm0.01}$ |
| OOD | Cross | 0.99 | 0.00 | 0.01 | 0.00 | $0.865_{\pm0.17}$ | 0.01 | 0.71 | 0.28 | $0.865_{\pm0.04}$ | 0.01 | 0.99 | $0.751_{\pm0.07}$ |
| | Div | 0.86 | 0.01 | 0.11 | 0.02 | $0.818_{\pm0.22}$ | 0.00 | 0.94 | 0.06 | $0.859_{\pm0.03}$ | 0.00 | 1.00 | $0.843_{\pm0.03}$ |
| | KL | 0.91 | 0.02 | 0.07 | 0.00 | $0.822_{\pm0.21}$ | 0.05 | 0.76 | 0.19 | $0.818_{\pm0.06}$ | 0.00 | 1.00 | $0.812_{\pm0.04}$ |
| | L1 | 0.91 | 0.00 | 0.08 | 0.01 | $0.813_{\pm0.20}$ | 0.02 | 0.62 | 0.36 | $0.847_{\pm0.07}$ | 0.14 | 0.86 | $0.724_{\pm0.11}$ |
| | L2 | 0.84 | 0.02 | 0.13 | 0.01 | $0.729_{\pm0.23}$ | 0.03 | 0.63 | 0.34 | $0.798_{\pm0.11}$ | 0.12 | 0.88 | $0.651_{\pm0.10}$ |
| **DiffDiv (ours)** | Cross | 0.96 | 0.00 | 0.04 | 0.00 | $0.856_{\pm0.16}$ | 0.00 | 0.94 | 0.06 | $0.836_{\pm0.05}$ | 0.01 | 0.99 | $0.745_{\pm0.10}$ |
| | Div | 0.94 | 0.00 | 0.06 | 0.00 | $0.916_{\pm0.13}$ | 0.00 | 0.98 | 0.02 | $0.826_{\pm0.05}$ | 0.00 | 1.00 | $0.857_{\pm0.01}$ |
| | KL | 0.89 | 0.01 | 0.10 | 0.00 | $0.786_{\pm0.20}$ | 0.00 | 0.94 | 0.06 | $0.837_{\pm0.05}$ | 0.04 | 0.96 | $0.672_{\pm0.07}$ |
| | L1 | 0.89 | 0.00 | 0.09 | 0.02 | $0.784_{\pm0.20}$ | 0.00 | 0.77 | 0.23 | $0.816_{\pm0.11}$ | 0.08 | 0.92 | $0.659_{\pm0.12}$ |
| | L2 | 0.93 | 0.02 | 0.03 | 0.02 | $0.762_{\pm0.22}$ | 0.01 | 0.83 | 0.16 | $0.757_{\pm0.12}$ | 0.09 | 0.91 | $0.650_{\pm0.11}$ |

cues 100% of the times, as models trained on UTKFace attend to *ethnicity* cues, showcasing strong preferential cue bias while achieving near-perfect classification on diagonal –*id*– validation data. Additionaly, we find similar behaviors for models trained on CelebA, which solely preferentially select the *pale skin* feature of *oval face* during training. Notably, upon introducing the diversification objectives, we observed a perceptible shift in the models' behavior, some of which averted their focus from the primary, easy-to-learn cues, turning instead to other latent cues present within the data. Among the objectives considered, $kl$, $L_1$, and $L_2$ exhibited the highest cue diversity, catalyzing the ensemble to distribute attention across multiple cues. However, this is at the expense of a marked drop in the average ensemble performance. Conversely, the *div* and *cross* objectives yielded milder diversification, focusing on the next readily discernible cues: *scale* in ColorDSprites, *gender* in UTKFace, and *oval face* in CelebA; while maintaining a generally higher ensemble validation performance. Similarly to Fig. 5, We find the diversification induced by DiffDiv in Table 2 is largely comparable with *ood* data on ColorDSprites, with respectively up to 11% and 14% of the models averting their attention from the main 'color' cue in ColorDSprites; up to 23% and 38% of the models averting their attention to the ethnicity cue in UTKFace; and up to 9% and 14% of the models averting their attention to the ethnicity cue in CelebA .

**Increased Ensemble Disagreement Negatively Correlates with Ensemble *id* Performance:** Achieving ensemble diversity by disagreement on unlabelled *ood* samples has previously been shown to negatively impact *id* ensemble performance (Pagliardini et al., 2022) (suppl. Fig. S2), while improving *ood* performance on downstream tasks associated with non-shortcut cues. Generally, ensemble model selection has been an effective method to prune models that misalign with the original classification objective (Lee et al., 2022). In this section, we wish to understand this relationship under the scope of Diffusion-guided diversification. In Fig. 6 we observe the change in ensemble average accuracy $\Delta_{acc}$ as a function of the change in diversity $\Delta_{ds}$, spanning all the diversification objectives. We highlight a few important observations. First, increased ensemble diversity via disagreement negatively correlates with average ensemble accuracy, without additional model selection mechanisms. Second, real *ood* data achieves the highest diversity gain, comparable to the diffusion samples at the *originative* stage (§4.2), $\approx [10, 100]$ for ColorDSprites, $\approx [450, 800]$ for UTKFace, and $\approx [1000, 1200]$ for CelebA. Lastly, the diversification objectives show comparable trends, with the $div$ objective displaying marginally better diversification/accuracy performance than the others on both tasks.

## 5    DISCUSSION

### SUMMARY

Shortcut learning is a phenomenon undermining the performance and utility of deep learning models, where easy-to-learn cues are preferentially learned for prediction, regardless of their relevance to the downstream task. We propose *DiffDiv*, a framework to achieve shortcut learning mitigation via ensemble diversification on low-fidelity DPM counterfactuals. We train and compare diverse ensembles on different datasets, disagreement objectives, and diversification conditions, and show several important findings.

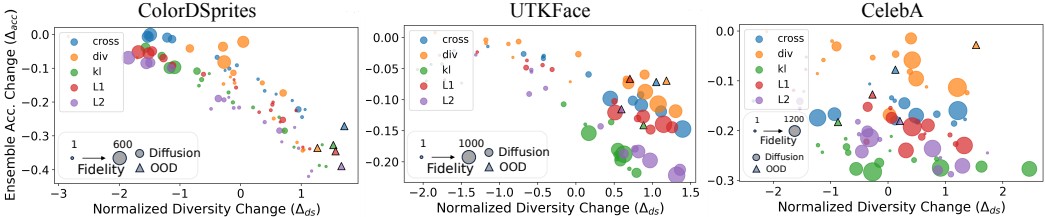

Figure 6: The relationship between the change in normalized classification prediction diversity ($\Delta_{ds}$) and the change in validation accuracy of the ensemble ($\Delta_{acc}$), when trained with samples from DPMs at varying levels of fidelities. The $\Delta$s are computed with respect to baseline ensemble training, with no diversification objective. We also compare the metrics achieved by diversification with non-correlated, off-diagonal, *ood* data from the respective datasets.

**DPM Training and Counterfactual Generation:** First, we show that DPMs can generate novel feature combinations even when trained on images displaying correlated cues. We observe this phenomenon as a function of diffusion training epochs. We identify three relevant different stages within DPM training, namely: *burn-in*, *originative*, and *exact* stage. Importantly, the *originative* stage suggests early tendencies of DPMs to learn the manifold of the distribution under scrutiny, without (over)fitting the intricate nuances of the training data, leaving space for the generation of samples with novel feature combinations even when trained on data presenting correlated features. We find that the low-fidelity, diverse, samples at this stage are especially useful for ensemble diversification.

**Diffusion Ensemble Diversification:** We show that diffusion-guided diversification leads models to avert attention from shortcut cues, and that diffusion counterfactuals can lead to comparable ensemble diversity without the need for expensive *ood* data collection. In our experiments, we consider several diversification objectives and find that our central hypothesis is true despite this choice. Moreover, we find a relationship between the level of diffusion fidelity and the effectiveness of ensemble diversification. In particular, we show that ensemble diversity and validation ID performance can be used as a proxy for the identification of the DPM *originative* stage.

LIMITATIONS, IMPLICATIONS AND FUTURE DIRECTIONS

**Beyond in-distribution sampling:** Although beyond the scope of this work, it would be worthwhile to understand the mechanism behind the ability of DPMs to generalize beyond the observed feature combinations even under feature correlation. An implication of our findings is the potential of early-stopping mechanisms to enforce these particular generative capabilities. Another significant implication is the potential of DPMs for feature disentanglement and *ood* sample generation, which may have interesting repercussions in several important domains including data augmentation and *ood* generalization.

**Diversification:** We believe it is worthwhile to delve deeper into the interplay between the fidelity of DPM-generated samples and model diversification. A limiting factor in the disagreement objective to achieve shortcut mitigation lies in its impact on average ensemble performance. In prior work, this was partly overcome by careful model selection, but new avenues should be explored to maintain *iid* performance while achieving diverse *ood* prediction via disagreement.

Furthermore, although our findings showcase an important phenomenon, and its applicative utility for diverse ensembles, it would also be valuable to extend this work to additional data modalities and explore its implications beyond vision.An interesting parallel venue is the use of text-to-image models for conterfactual generation Dunlap et al. (2023); Howard et al. (2023) which may be especially helpful when prior knowledge of the biases is known.

## 6 CONCLUSION

This work presents a step forward in addressing the challenge shortcut learning in DNNs. By leveraging the unique capabilities of DPMs for ensemble diversification, we provide a practical method that achieves shortcut mitigation in a variety of visual tasks, with only negligible performance loss compared to methods requiring expensive *ood* data collection.

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

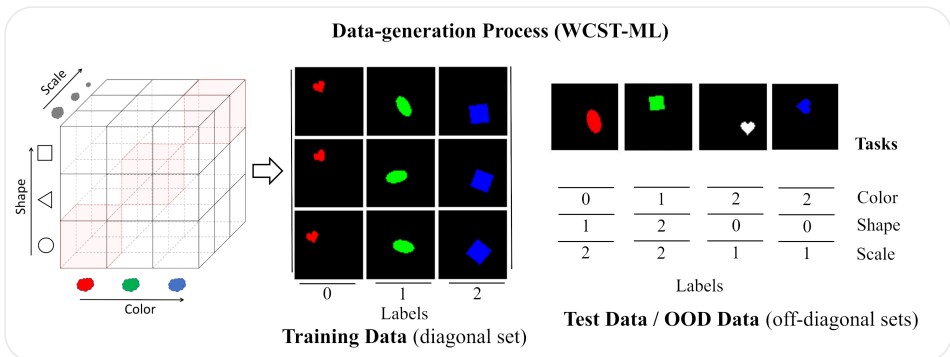

Figure S1: We consider the WCST-ML setting (Scimeca et al., 2022) as the experimental ground in our experiments. Each dataset is partitioned into training data where the task labels are perfectly correlated with the image input features (e.g. {color, shape, and scale}) and Test data, where samples are used to test a model's tendency to a feature over another. We use a random subset of Test Data for OOD experiments.

## S1 Supplementary Methods

### S1.1 Datasets

In this work, we leverage three representative datasets: a color-augmented version of DSprites (Matthey et al., 2017), UTKFace (Zhang et al., 2017), and CelebA (Liu et al., 2015).

**DSprites:** DSprites includes a comprehensive set of symbolic objects generated with variations in five latent variables: shape, scale, orientation, and X-Y position. We augment this dataset with a color dimension and remove redundant samples (e.g. due to rotations), resulting in $2,862,824$ distinct images, which we refer to as ColorDSprites. ColorDSprites permits the examination of shortcut biases in a highly controlled setup.

**UTKFace:** UTKFace provides a dataset of $23,708$ facial images annotated with attributes like age, gender, and ethnicity. Unlike DSprites, UTKFace presents a real-world, less controlled setup to study bias. Its inherent complexity and diversity make it an ideal candidate for understanding the model's cue preferences when societal and ethical concerns are at stake.

**CelebA:** CelebA is a large-scale dataset comprising $202,599$ celebrity facial images annotated with 40 different attributes, including gender, age, and facial features. It provides a more extensive and diverse real-world dataset compared to UTKFace, offering rich variations in pose, background, and lighting. CelebA is commonly used for studying bias and fairness in models due to its attribute diversity and challenging conditions.

### S1.2 Operationalizing WCST-ML Across Datasets

We follow the set-up in (Scimeca et al., 2022) and construct a balanced dataset $\mathcal{D}_{\text{diag}}$, which includes a balanced distribution of cues, coupled with their corresponding off-diagonal test sets (one for each feature) Figure S1. For both datasets, we define a balanced number of classes $L$ for each feature under investigation. Where the number of feature values exceeds $L$, we dynamically choose ranges to maintain sample balance with respect to each new feature class. For instance, for the continuous feature 'age' in UTKFace, we dynamically select age intervals to ensure the same $L$ number of categories as other classes, as well as sample balance within each category. We consider sets of features previously found to lead to strong simplicity biases. For ColorDSprites, we consider $K_{DS} = 4$, features $\{color, orientation, scale, shape\}$, and $L = 3$ as constrained by the number of shapes in the dataset. Within UTKFace we consider $K_{UTK} = 3$, features $\{ethnicity, gender, age\}$, and $L = 2$ as constrained by the binary classification on *gender*. For CelebA we consider $K_{CL} = 2$, features $\{lightskin, ovalface\}$, and $L = 2$ as enforced by the binary labels on all features. For each

dataset we create one *diagonal* subset of fully correlated features and labels, available at training time, and $K_{DS}$, $K_{UTK}$ and $K_{CL}$ feature-specific *off-diagonal* datasets to serve for testing the models' shortcut bias tendencies.

### S1.3 DPM Training and Synthetic Counterfactual Generation

We utilize Diffusion Probabilistic Models (DPMs) to generate synthetic data for our experiments. DPMs operate by iteratively adding or removing noise from an initial data point $x$ through a stochastic process governed by a predefined noise schedule. We base our training regime on (Ho et al., 2020). As denoiser, we train a classic U-Net architecture with 4 down-sampling blocks and 4 up-sampling blocks with $\approx 9mil$ parameters for all datasets. We train the model with the objective in Equation 3 by iterating through the relevant dataset over a maximum of 1200 epochs. We use a vanilla Adam optimizer in all experiments. All DPM schemes use a time discretization of 1000 steps. To facilitate efficient sampling, we employ Denoising Diffusion Implicit Models (DDIM) (Song et al., 2020), a first-order ODE solver for DPMs (Salimans & Ho, 2022; Lu et al., 2022), utilizing a predictor-corrector scheme to minimize the number of sampling steps, lowering the final number to 250 during sampling in framework.

For each of the experiments, we use the trained DPM to generate a fixed dataset of 3000 synthetic counterfactuals, which is independently shuffled and batched-sampled for the diversification objective during ensemble training. We perform ablation studies considering a larger batch of synthetic counterfactuals (equal to the number of data points in each dataset) in §S2.5, with only marginal performance gains compared to the smaller set.

## S2 Supplementary Results

### S2.1 γ Selection

We perform a hyper-parameter search to find the values of $\gamma$ for each diversification objective. We perform the search via the same methodology used in the *ood* main experiments, i.e. by training an ensemble of 100 ResNet-18 models on the fully correlated *diagonal* datasets, while diversifying a small subset of 30% of the original training data, randomly sampled from the de-correlated left-out set. We consider $\gamma$ values ranging from $1e-3$ to $1e1$, and monitor both the validation ensemble accuracy as well as the predictive diversity on a separate de-correlated set of validation data. Figure S2 shows the performance of each metric for different values of $\gamma$. We select the values reported in Table S1 to be at the intersection of the accuracy and diversification trends for each model.

### S2.2 Model Selection to Boost Diverse Ensemble Performance

The mitigation of Shorcut leaning through diversification is generally known in the literature to suffer from a decrease in ensemble ID performance, as we also observe and study in our experiments. A prominent methodology to mitigate this phenomenon in the literature is ensemble model selection, where a subset of models is selected for final ensemble inference. We perform additional experiments to assess the degree by which model selection can aid in ensemble performance within the DiffDiv framework. To ensure diversity in the final selection, we include in the selected subset any model showing shortcut-cue aversion from Table 1. Furthermore, we select additional models to reach a dynamic range between 15% and 99% of the original ensemble. We compare the performance of the ensemble before and after model selection in Table S2.

### S2.3 On the Influence of DPM Fidelity to Diversification

We perform experiments whereby an ensemble of 100 ResNet-18 models is trained separately with respect to all diversification objectives considered. Figure S3 shows the diversification results as a function of the fidelity of the DPM used to sample the diversification set. Although we observe that the diversification level obtained is dependent on the diffusion fidelity level, suggesting the need for appropriate early stopping procedures to achieve increased ensemble prediction diversity, we find DiffDiv to not be overly sensitive to this choice. In fact, we observe broad areas with similar diversity levels across several DPM fidelities. While the trends vary mildly across different disagreement

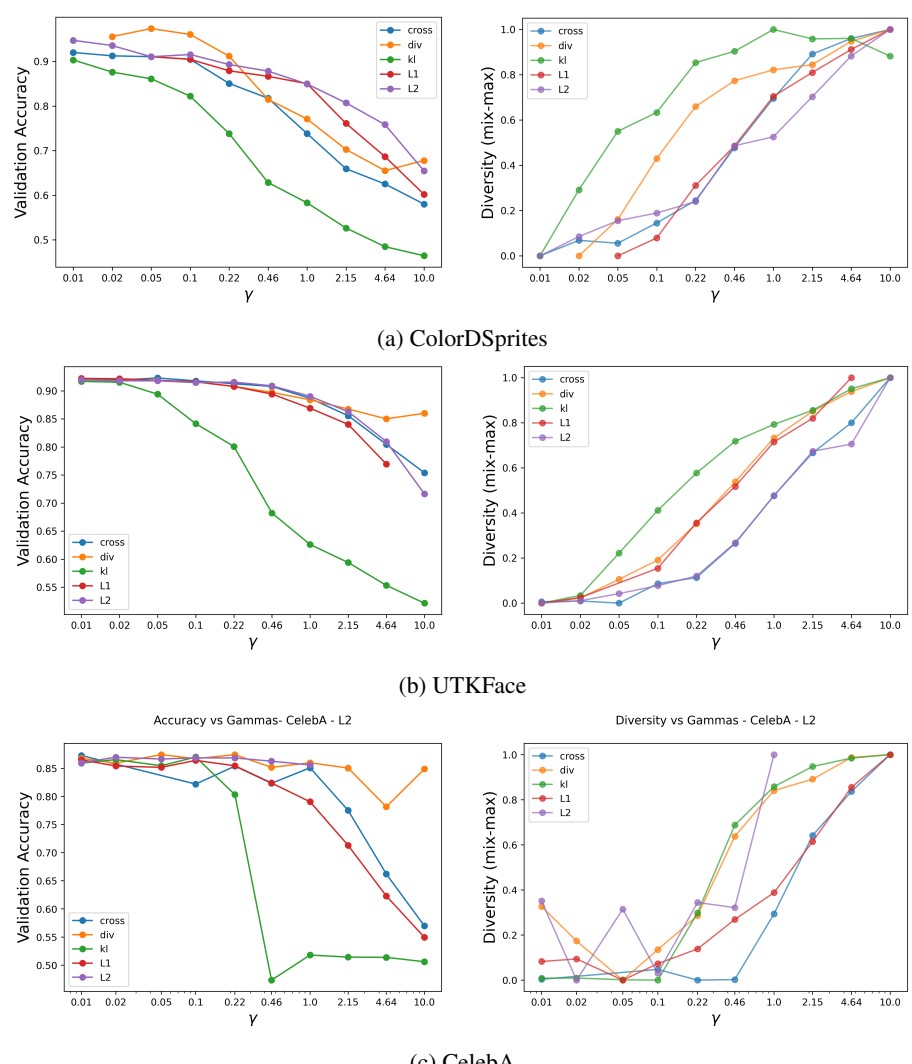

(a) ColorDSprites

(b) UTKFace

(c) CelebA

Figure S2: Hyper-parameter search on the disagreement intensity ($\gamma$) for each diversification objective. For the main experiments, we select the values of $\gamma$ at the intersection of the accuracy-diversity trends by each objective (Table S1).

objectives, we find the diversification maximized in ColorDSprites and UTKFace coherently with the analysis in Section 4.2, where improved diversification is achieved around the *originative* interval. For ColorDSprites, this is realized in around 20 DPM training epochs, for UTKFace, it is realized in around 800 DPM training epochs, while for CelebA it is around 1000 DPM training epochs.

Interestingly, our results suggest how ensemble diversification metrics can be a viable proxy for appropriate *originative* DPM training. In Fig. S4, we observe the min-maxed change in accuracy and change in diversity by the ensembles with respect to baseline training on ColorDSprites. In the figure, we observe similar trends across all diversification methods, whereby the *originative* interval (highlighted in gray) is primarily identified by the highest changes in diversity, while also considering the least drop in accuracy. These results confirm our previous supervised findings (Fig. 3). Imporantly, we find that only when jointly looking at both diversity and validation performance we can best identify the relevant areas around the generative stage. Under this light, ensemble performance/diversity validation metrics can be directly leveraged for DPM early stopping logic. Our results align with our previous observations, where the highest number of ood-generated samples to lie within the DPM training intervals achieve the highest change in diversity while maintaining good classification performance (Fig. 6.)

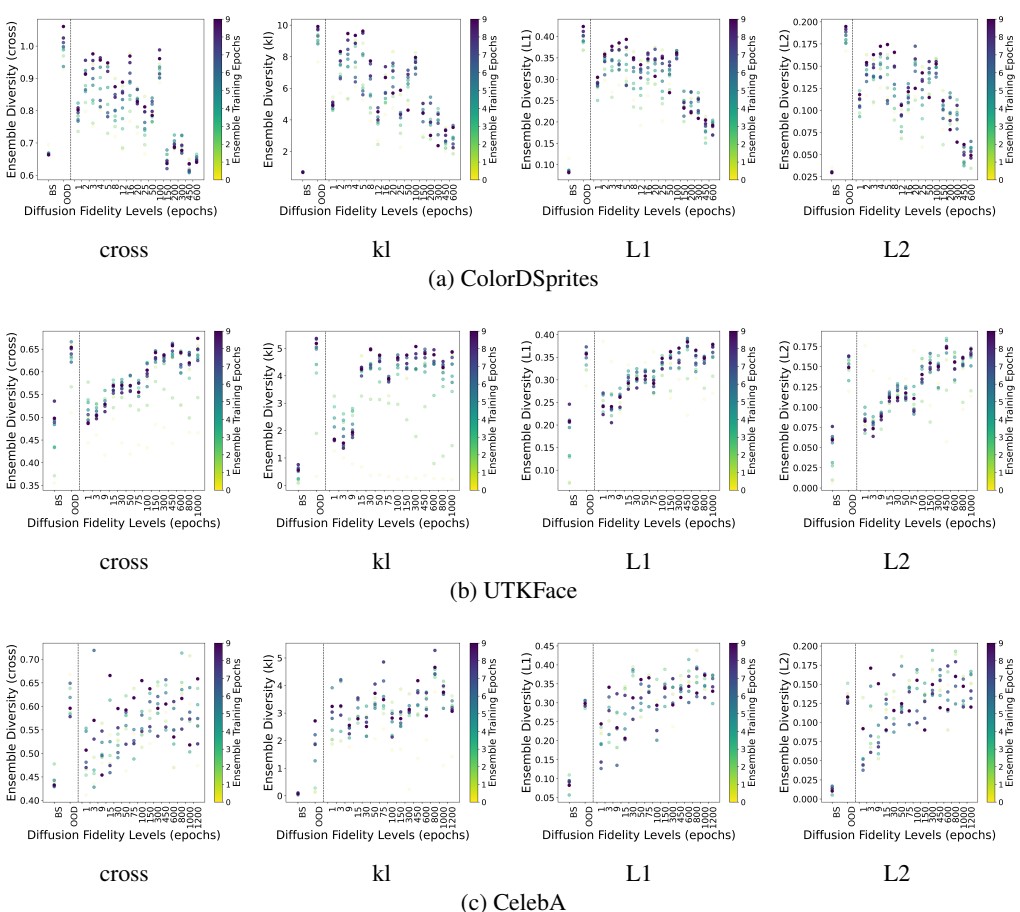

Figure S3: Ensemble diversity as enforced via samples from diffusion models trained a different fidelity levels (i.e. diffusion training epochs).

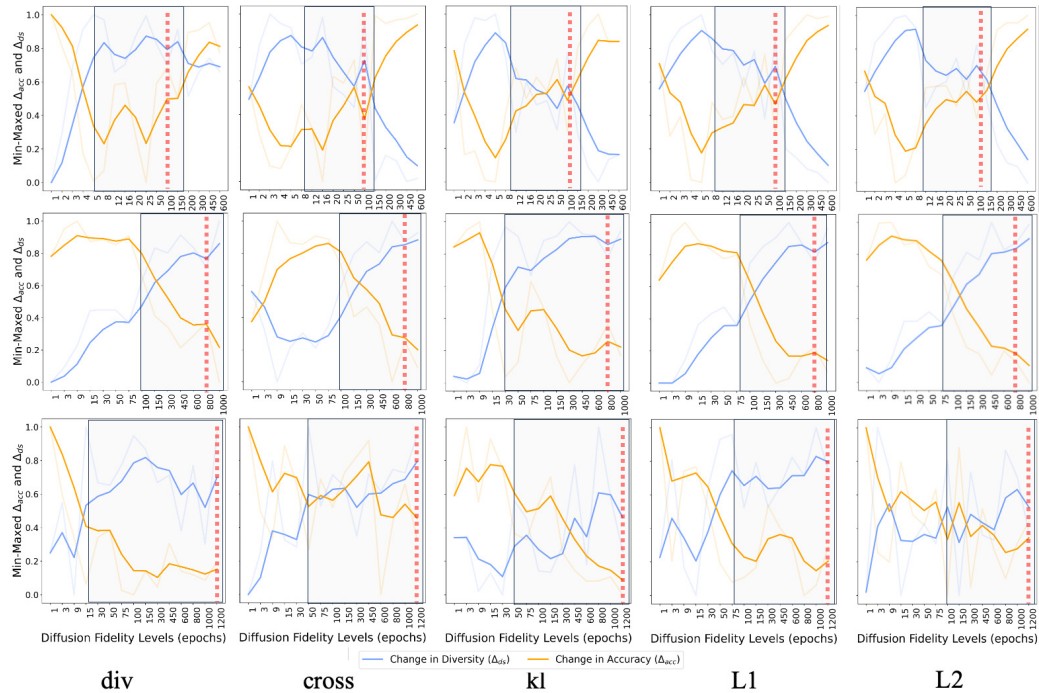

Figure S4: Min-Maxed change in accuracy and diversity by ensembles trained with diffusion-augmented samples on ColorDSprites, with respect to all considered diversification methods. The *originative* stage, as qualitatively identified in the experiments (see Fig. 3 and Fig. 4) is shown in gray. The areas primarily highest in diversity, with the least change in accuracy, are significant of the originative stage. We mark with red discontinuous vertical bars the stage used for the experiments

Table S1: Disagreement $\gamma$ used in our experiments.

|  | L2 | L1 | cross | div | kl |
|---|---|---|---|---|---|
| **ColorDSprites** | 5.0 | 1.5 | 0.5 | 0.5 | 0.1 |
| **UTKFace** | 5.0 | 2.0 | 2.0 | 5.0 | 0.2 |
| **CelebA** | 5.0 | 2.0 | 2.0 | 5.0 | 0.2 |

## S2.4 DIVERSIFICATION LEADS TO ENSEMBLE MODELS ATTENDING TO DIFFERENT CUES

Figure S5 illustrates a feature-centric description of 10 ensemble models trained with a diversification objective on *ood* data (a) and Diffusion generated counterfactuals (b). The variation across models is evident: several models substantially reduce their dependency on the leading cues of the respective datasets (black edges), diverging considerably from the almost identical configurations present in the baseline ensemble (red edges). For some of the models, in fact, the averted attention on the main

Table S2: Comparison between the ensemble accuracy before (Ens) and after model selection (Select). Each subset of models in the ensemble is chosen dynamically while keeping all models with cue averted tendencies in Table 2

| Dataset → | ColorDSprites | | UTKFace | | CelebA | |
|---|---|---|---|---|---|---|
| Obj. ↓ | Valid. Acc. (Ens) | Valid. Acc. (Select) | Valid. Acc. (Ens) | Valid. Acc. (Select) | Valid. Acc. (Ens) | Valid. Acc. (Select) |
| Baseline | $1.000 \pm 0.00$ | $1.000 \pm 0.00$ | $0.920 \pm 0.02$ | $0.943 \pm 0.00$ | $0.857 \pm 0.01$ | $0.873 \pm 0.00$ |
| Cross | $0.856 \pm 0.16$ | $0.945 \pm 0.14$ | $0.836 \pm 0.05$ | $0.856 \pm 0.07$ | $0.745 \pm 0.10$ | $0.828 \pm 0.06$ |
| Div | $0.916 \pm 0.13$ | $0.980 \pm 0.07$ | $0.826 \pm 0.05$ | $0.868 \pm 0.06$ | $0.857 \pm 0.01$ | $0.873 \pm 0.00$ |
| KL | $0.786 \pm 0.20$ | $0.872 \pm 0.22$ | $0.837 \pm 0.06$ | $0.858 \pm 0.08$ | $0.672 \pm 0.07$ | $0.713 \pm 0.09$ |
| L1 | $0.784 \pm 0.20$ | $0.861 \pm 0.23$ | $0.816 \pm 0.11$ | $0.824 \pm 0.11$ | $0.659 \pm 0.12$ | $0.737 \pm 0.12$ |
| L2 | $0.762 \pm 0.22$ | $0.864 \pm 0.26$ | $0.757 \pm 0.12$ | $0.776 \pm 0.13$ | $0.650 \pm 0.11$ | $0.716 \pm 0.12$ |

Table S3: Diversification results on ColorDSprites, UTKFace, and CelebA when using the same number of *ood* samples as the training dataset. The feature columns report the fraction of models (in each row) biased towards the feature. The final column reports the average validation accuracy for the ensemble when tested on a left-out feature-correlated *diagonal* set, of the same distribution as the original training data

| Dataset → | ColorDSprites | | | | | UTKFace | | | | CelebA | | |
|---|---|---|---|---|---|---|---|---|---|---|---|---|
| Obj. ↓ | Color (↓) | Orient. | Scale | Shape | Acc. (↑) | Age | Ethnicity (↓) | Gender | Acc. (↑) | Oval Face | Pale Skin (↓) | Acc. (↑) |
| Baseline | 1.00 | 0.00 | 0.00 | 0.00 | $1.000_{\pm 0.00}$ | 0.00 | 1.00 | 0.00 | $0.920_{\pm 0.02}$ | 0.00 | 1.00 | $0.857_{\pm 0.01}$ |
| Cross | 0.92 | 0.00 | 0.08 | 0.00 | $0.849_{\pm 0.16}$ | 0.00 | 0.73 | 0.27 | $0.858_{\pm 0.04}$ | 0.01 | 0.99 | $0.751_{\pm 0.07}$ |
| Div | 0.82 | 0.00 | 0.18 | 0.00 | $0.820_{\pm 0.19}$ | 0.00 | 0.76 | 0.24 | $0.844_{\pm 0.04}$ | 0.00 | 1.00 | $0.843_{\pm 0.03}$ |
| KL | 0.90 | 0.03 | 0.05 | 0.02 | $0.801_{\pm 0.20}$ | 0.02 | 0.68 | 0.30 | $0.820_{\pm 0.07}$ | 0.00 | 1.00 | $0.812_{\pm 0.04}$ |
| L1 | 0.90 | 0.01 | 0.07 | 0.02 | $0.799_{\pm 0.21}$ | 0.00 | 0.66 | 0.34 | $0.832_{\pm 0.09}$ | 0.14 | 0.86 | $0.724_{\pm 0.11}$ |
| L2 | 0.86 | 0.04 | 0.08 | 0.02 | $0.745_{\pm 0.22}$ | 0.08 | 0.58 | 0.34 | $0.761_{\pm 0.15}$ | 0.12 | 0.88 | $0.651_{\pm 0.10}$ |

shortcut cue leads to increased reliance on one of the other observed features (e.g. scale and age for models 7 (ColorDsprites), 73 (UTKFace) and 5 (CelebA) in Fig. S5a, and models 80 (ColorDsprites), 47 (UTKFace) and 61 (CelebA)) in Fig. S5b).

## S2.5 On the Influence of an Increased Number of *ood* Samples for Ensemble Disagreement

As per the original objective, with DPM sampling we aim to circumvent the diversification dependency on Out-Of-Distribution data, which is often not readily accessible and can be costly to procure. We test this dependency further and assess the quality of the diversification results when matching the number of *ood* data used for diversification to the original training data for the ensemble. We report in Table S3 our findings. We observe the quality of the disagreement on ColorDSprites to only marginally benefit from additional disagreement samples, with approximately $1\%$ to $7\%$ more of the models to avert their attention from the shortcut cue *color* as compared to the original experiments. On the other hand, we observe a strong improvement in the diversification for UTKFace, mainly registered via the *div* objective, where $24\%$ of the models averted their attention from the *ethnicity* shortcut, as opposed to the original $6\%$ in our previous experiments, while maintaining high predictive performance on the validation set. We observe marginal improvements on the other objectives, with approximately $4\%$ to $8\%$ additional models achieving cue aversion. We speculate this gain to be due to the higher complexity of the features within the data, which may require additional specimens for appropriate diversification.

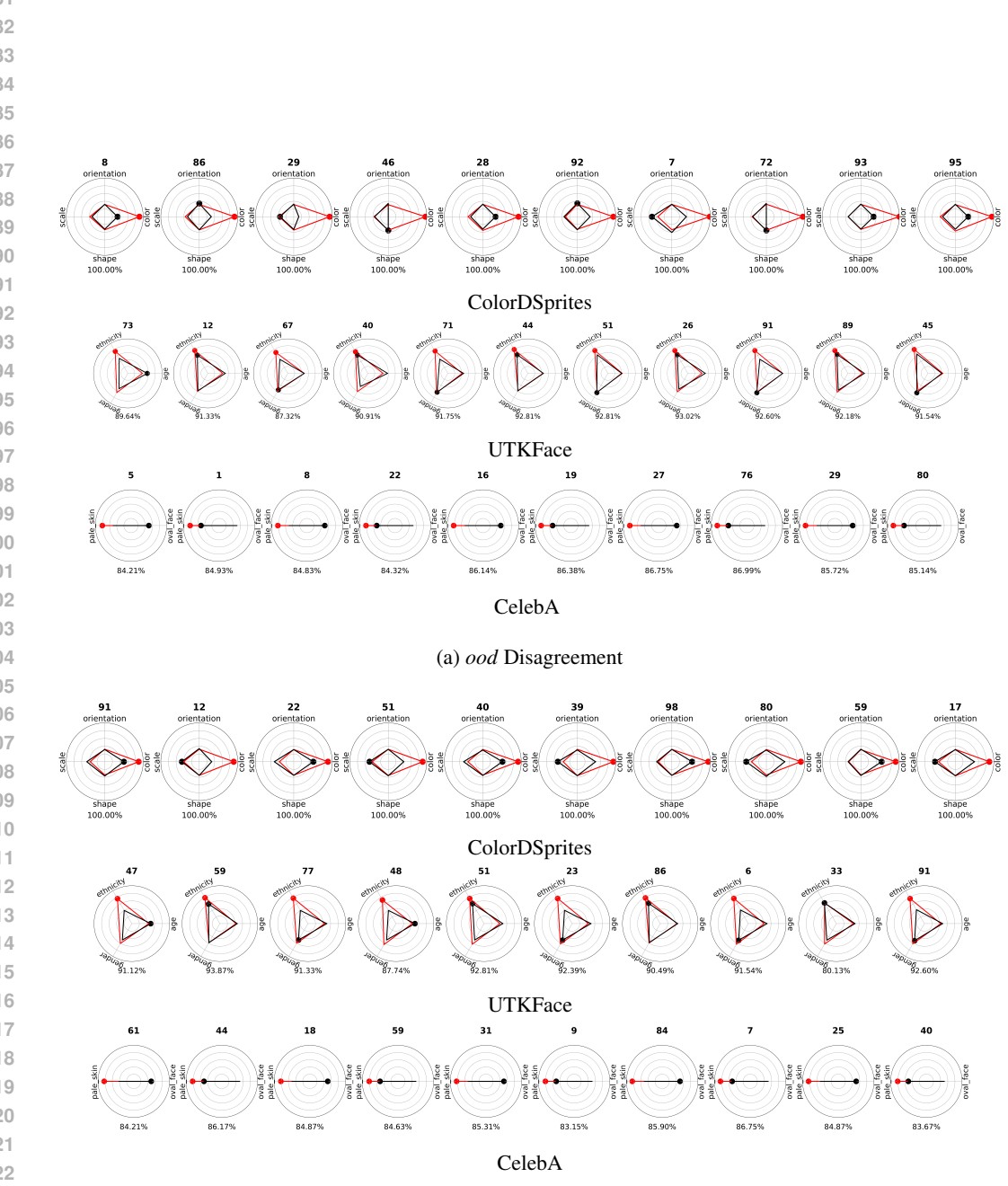

(a) *ood* Disagreement

(b) Diffusion Disagreement

Figure S5: Comparison of 10 diversified models when training the ensemble while using (a) feature-uncorrelated *ood* data and (b) Diffusion Samples.

