# OpenReview forum: "Mitigating Shortcut Learning with Diffusion Counterfactuals and Diverse Ensembles"
_ICLR.cc/2025/Conference — Submitted to ICLR 2025_

### Official Review · Reviewer_GGVZ · 2024-11-04

**Soundness:** 3
**Presentation:** 3
**Contribution:** 3
**Rating:** 6
**Confidence:** 4

**Summary:**

The paper presents a method using Diffusion models for diversifying sets of ensemble models. The authors use Diffusion models to generate synthetic training data that presents novel feature combinations, even when the original training data has highly correlated features, providing a representation of the original data that covers a broader set of features and hides potentially dangerous short cut cues. The authors demonstrate that using their synthetic data for ensemble training helps improve model generalization to out-of-distribution samples, diversity, and mitigation of shortcut bias.

**Strengths:**

The paper shows originality by proposing a novel approach to address shortcut learning with DPMs to generate synthetic counterfactuals for ensemble diversification.

The application of DPMs for mitigating bias without relying on labeled out-of-distribution data is particularly innovative. Through the use of DPMs to generate feature combinations unseen in the training data, the research extends beyond conventional methods that depend on auxiliary datasets or explicit feature labels.

The authors use a few challenging datasets to demonstrate the efficacy of their method.

Writing is clear with ample citations provided should it be difficult to follow.

The paper content has significant practical applications for ML practitioners dealing with entangled factors in their training data.

**Weaknesses:**

The paper could benefit from more comparisons to competing methods, even if those methods require out-of-distribution data as an assumption without any mechanism in place for data/counterfactual generation.

More discussion on the advantages of using DPMs for counterfactual generation could be beneficial, particularly some discussion on the generation flexibility of DPMs and how they handle feature disentanglement.

Despite providing a reasonable method, some of the “work” being done is obfuscated behind the use of DPMs and lack of direct comparisons to other methods.

**Questions:**

1. Are there more competing methods? Please compare with the state-of-the-art methods. Otherwise, there is no way to assess the proposed method accurately.

2. Have you considered applying the DiffDiv framework beyond computer vision, such as in text or structured data? If so, what challenges or modifications do you anticipate for adapting it to non-visual data?

3. How do you determine the optimal training stage (e.g., burn-in, originative, exact) for DPMs to generate counterfactuals effectively across different datasets? Could this be automated for practical use? Is there a way to systematically balance sample fidelity and the diversity required for effective training?

---

> ### Author Response · Authors · 2024-11-19
> **Thank you + W1 & Q1 (a) and (b) Comparison with current methods**
>
> We sincerely thank the reviewer for their thoughtful and thorough feedback, as well as for recognizing the novelty and practical significance of our approach using Diffusion Models (DPMs) for ensemble diversification and bias mitigation. We have carefully considered your suggestions and made several changes to strengthen the manuscript, detailed below in our responses.
>
> ## **W1 & Q1) Comparison with current methods and accurate assessment of DiffDiv**
>
> We break down our answer into three parts, to precisely address the reviewer's point. Please find (a) and (b) below
>
> #### **a) Comparison with other diversification methods**
> Thank you for raising this point. In the original manuscript, we did not explicitly clarify how we incorporate and validate past research in our work. We indeed benchmark state-of-the-art diversification methods including recent state-of-the-art diversification objectives, such as from DivDis [1] and Agree2Disagree [2]. Furthermore, these diversification methods are fully compatible with our DiffDiv framework, and we show how they can benefit from using DPM counterfactuals rather than OOD data. On the other hand, we show how DiffDiv works despite the choice of the method for diversification.
>
> Following the reviewer's comment we have revised the results section to clarify these aspects. We have also revised Table 2 to better highlight the results with respect to each benchmarked diversification methodology.
>
> #### **b) Comparison with other bias mitigation methods**
> In this work, we use WSCT-ML [3] as the principal framework for data generation and analysis through which we observe the experiments, since it provides a clear method to partition the data and observe the downstream efficacy of shortcut mitigation with respect to the cues in the data. This method fundamentally differs from others in the sense that there is "no right cue" to use for classification. Instead, each cue is in itself a different task, and a fully correlated dataset exposes the bias tendencies of models. Notably, improving on a particular cue -- e.g. shape-- will unavoidably worsen performance on other cues in WSCT-ML.
>
> This assumption generally breaks most current methods for shortcut mitigation. We direct the reviewer to the answer **W1 & Q2 (b)** and **\(c\)** to reviewer **Fptv** for a tangential discussion of text-to-image methods for this purpose. As DiffDiv hinges primarily upon diversification as the means to achieve shortcut mitigation, we instead benchmark and compare prominent diversification methods in the literature, as better explained in part **(a)**.
>
>
> **References**
>
> [1] Lee, Y., Yao, H. and Finn, C., 2022. Diversify and disambiguate: Learning from underspecified data. arXiv preprint arXiv:2202.03418.
>
> [2] Pagliardini, M., Jaggi, M., Fleuret, F. and Karimireddy, S.P., 2022. Agree to disagree: Diversity through disagreement for better transferability. arXiv preprint arXiv:2202.04414.
>
> [3] Scimeca, L., Oh, S.J., Chun, S., Poli, M. and Yun, S., 2021. Which shortcut cues will dnns choose? a study from the parameter-space perspective. arXiv preprint arXiv:2110.03095.

---

> ### Author Response · Authors · 2024-11-19
> **W1 & Q1 (c) Additional results for the accurate assessment of DiffDiv**
>
> We extend our original results and analysis to test even further how DiffDiv can lead to bias mitigation and a performance boost over ensemble training. We add two key experiments:
>
> Firstly, we compute and report the average increase in accuracy for each of the benchmarked diversification methods over the non-shortcut cues in DiffDiv, ensuring that the bias mitigation indeed leads to a more robust feature set for prediction. We summarize the results in the table below:
>
>
> | Obj. ↓ Dataset → | Color | Face  | CelebA |
> |--------------------|-------|-------|--------|
> | Cross              | 5.17  | 3.37  | 2.00   |
> | Dis                | 8.16  | 3.61  | 0.45   |
> | KL                 | 2.87  | 4.54  | 2.02   |
> | L1                 | 3.31  | 3.61  | 1.30   |
> | L2                 | 4.67  | 5.04  | 1.34   |
>
> We observe an always positive change in accuracy over the averted cues, extending the diversification and cue aversion results in Table 2. We have included these results as a new table in the manuscript (Table 1).
>
> Secondly, inspired by [1], we have performed ablation studies on model selection post-training. We include in the selected subset any model showing shortcut-cue aversion from Table 2. We then select an additional set of models to reach a dynamic range between 15% and 99% of the original ensemble. We show below the performance of the ensemble after model selection (as opposed to the original performance in parenthesis), and observe a severe performance increase with respect to our baseline results.
>
> | **Obj.↓**   Dataset→     |  Model Select. Acc. (Ensemble Acc.) <ColorDSprites> |  Model Select. Acc. (Ensemble Acc.) <UTKFace> | Model Select. Acc. (Ensemble Acc.) <CelebA>|
> |-----------------|-----------------------------------|-----------------------------------|-----------------------------------|
> | Baseline        | 1.000                      | 0.943                     | 0.873                      |
> | Cross           | 0.945 (0.856)                     | 0.856 (0.836)                     | 0.828 (0.745)                     |
> | Dis             | 0.980 (0.916)                     | 0.868 (0.826)                     | 0.873 (0.857)                     |
> | KL              | 0.872 (0.786)                     | 0.858 (0.837)                     | 0.713 (0.672)                     |
> | Std             | 0.861 (0.784)                     | 0.824 (0.816)                     | 0.737 (0.659)                     |
> | Var             | 0.864 (0.762)                     | 0.776 (0.757)                     | 0.716 (0.650)                     |
>
> Jointly, these results strengthen our claims further, demonstrating, in-depth, how DiffDiv can be a practical method to achieve significant bias mitigation.
>
> We have revised the manuscript to add these tables and results. All changes are highlighted in red.
>
> **References**
>
> [1] Lee, Y., Yao, H. and Finn, C., 2022. Diversify and disambiguate: Learning from underspecified data. arXiv preprint arXiv:2202.03418.

---

> ### Author Response · Authors · 2024-11-19
> **W2 & W3) Clarifications on DPM Training and counterfactual generation**
>
> Thank you for this comment. We have added a section in the supplementary materials providing further details on the denoising architecture used, diffusion training, and counterfactual generation (now Suppl. Methods Section S1.3).  We have revised the methods section, discussion, and supplementary materials to also more broadly discuss the role of DPMs in DiffDiv and their benefits in regard to the OOD phenomena we leverage. All changes are marked in red in the manuscript.

---

> ### Author Response · Authors · 2024-11-19
> **Q2) Data modalities beyond vision**
>
> This is a very interesting question! Indeed, we have considered the exploration of data modalities other than vision, including language and speech. We find this to be an exciting future research direction. In these, however, the community has yet to investigate in depth the inherent ability of DPMs to generate samples beyond the feature combinations observed in the original data distribution. In language, in particular, discrete diffusion may pose additional challenges, although, in principle, if diversification for shortcut mitigation is applicable, and the inherent ability of DPMs to generate outside of the trained distribution is true, then DiffDiv is indeed a viable method for bias mitigation. We have revised our discussion section to mention these future directions.

---

> ### Author Response · Authors · 2024-11-19
> **Q3 & W3) How to determine the optimal DPM training stage**
>
> Thank you for mentioning this. Firstly, It is important for us to mention that although the DPM training fidelity is indeed very important for appropriate diversification,  we observed that DiffDiv's performance is not highly sensitive to this choice. For example, we highlight the range of maximum diversity achieved through diversification in Figure 4 and Supplementary Figure S3 (ColorDSprites Fidelity 4-150; UTKFace Fidelity 400-1000; CelebA 50-1200). Notably, strong diversification occurs within approximately one-third of all DPM training steps for most datasets. Generally, we find that DiffDiv converges unless the diffusion model is nearly untrained, or overfit to the distribution (resulting in predominantly in-distribution (ID) samples) (see Figure 6).
>
> Secondly, as the reviewer mentions, it is indeed possible to automate DPM early stopping. We can identify the originative stage by using the ensemble as a proxy. In our experiments, we find the generative stage to be uniquely denoted by the highest change in ensemble diversity while keeping the negative change in accuracy to a minimum. These areas are generally broad and we observe similar DiffDiv performance across all relevant Fidelity levels.
>
> We clarify this important aspect in the manuscript and significantly revise and expand Supplementary Section 2.2. We also revise Figure S3 to show the change in accuracy and diversity by the ensembles trained with DPM samples at different training stages. Finally, we include in the figure the originative search experiments for all datasets. We highlight the areas displaying 'originative' tendencies and the fidelity level chosen for the experiments.

---

> > ### Comment · Reviewer_GGVZ · 2024-11-26
> >
> > Thanks for addressing the comments with so much detail. The paper will have a better quality after these improvements.

---

> ### Author Response · Authors · 2024-11-27
>
> Thank you again for your feedback and for raising pertinent and actionable questions. If any of our responses were satisfactory we would be extremely grateful if you could consider increasing your score. Please feel free to ask additional questions, we would be more than happy to provide further clarifications.

---

### Official Review · Reviewer_Fptv · 2024-11-04

**Soundness:** 3
**Presentation:** 3
**Contribution:** 2
**Rating:** 5
**Confidence:** 4

**Summary:**

The paper proposes a framework, DiffDiv, to mitigate spurious correlations in the data. The framework makes use of DPMs and the authors show that at certain training intervals, even if the model was trained on data with correlated data, it can generate novel feature combinations. Thus, counterfactual data can be generated and used to encourage ensemble diversification.

**Strengths:**

- The paper was well written and easy to follow.
- Analysis of diversity from DiffDiv was thorough and insightful.
- The idea of using diffusion models for diversification seems promising.

**Weaknesses:**

- The framework involves training a DPM which can be expensive for larger datasets. There may be cheaper ways to generate diverse data. E.g., ALIA [1] captions the training images and uses a language model to generate diverse prompts for a text to image. It involves several stages but does not require training a DPM. Furthermore, one of their experiments involves generating data to mitigate spurious correlations. It would also be interesting to compare against generating data from standard augmentations.
- Finding the best DPM epoch for generation in practice seems tricky. Furthermore, the originative interval sometimes leads to generations where the class is ambiguous (e.g., fig 3).
- Most of the analysis seems to be focused on showing the diversity of the data from DiffDiv. It would be useful to compare the downstream performance from using different types of additional data e.g., from augmentations or using pre-trained diffusion models to edit the training data.

[1] Dunlap et al. Diversify Your Vision Datasets with Automatic Diffusion-Based Augmentation.

**Questions:**

- How should the generation epoch be chosen in practice during training time?
- The generalization to different feature combinations of DPMs is not well understood, as mentioned in the limitations section. Thus, given the option of a framework like [1], that is less likely to fail silently, i.e., where we can check from the LLM summary if all features are extracted, what is the advantage of using DiffDiv?

---

> ### Author Response · Authors · 2024-11-19
> **Thank you + W1 & Q2 (a) DPM training cost**
>
> We wish to thank the reviewer for carefully reviewing our work, and raising questions leading to interesting discussion and improvements of our manuscript. We have made several changes to the manuscript to address your questions and concerns. Please find our responses below:
>
>  ###  **W1 \& Q2 DPM training cost, text-to-image models, and other data augmentation methodologies**
>
> To precisely address the reviewer's question we provide a three-part answer:
>
> #### **W1 \& Q2 (a)  DPM training cost**
> As the reviewer noted, DPM training and generation can be resource-intensive. However, in our framework, DPM training represents an amortized cost incurred only once. Additionally, due to the algorithm's characteristics, training is often stopped well before full convergence. In our experiments, we trained DPMs on DSprites for approximately 10-100 epochs (compared to 10k+ typically) and on UTKFace and CelebA for approximately 100-1200 epochs (versus 500k+ in literature). Generating counterfactuals also incurs minimal cost, as we generate a small set of 3,000 synthetic samples for all experiments, validated as sufficient to provide effective diversification cues (as supported by ablation experiments in Supplementary Section S2.5 and Table S3).
>
> Motivated by the reviewer’s comment, we have added an additional section in the supplementary materials detailing the denoising architecture, diffusion training, and counterfactual generation (now Supplementary Methods Section 1.3).

---

> ### Author Response · Authors · 2024-11-19
> **W1 & Q2 (b) Text-to-image models and why use DiffDiv vs a framework such as ALIA**
>
> Thank you for raising this interesting point and for providing a reference. Indeed, text-to-image models for data augmentation are highly relevant for diversification. Another notable concurrent work is [1], which utilizes stable diffusion for counterfactual generation to mitigate biases.
>
> A key distinction between these approaches lies in their underlying assumptions. For example, prior knowledge of specific shortcuts is often necessary to effectively generate a diverse set of samples for diversification through prompting. In [1], biases related to gender or race can be addressed, but other types of shortcut biases may be overlooked or even inadvertently reinforced due to prompt design. Additionally, text-to-image models have inherent limitations stemming from their shared embeddings; the diversity of generated samples is constrained by the quality and specificity of prompts. Finally, a perfectly trained text-to-image model will often struggle to generate images prompted to be outside its training domain [2], making it especially challenging to tackle non-canonical datasets. While large pre-trained models can mitigate some of these issues, they perform best when the data distribution closely resembles or is a subset of the model's training distribution. Significant distribution mismatches can cause issues similar to those seen during the "burn-in" phase of our experiments.
>
> In general, both methods have their advantages and limitations. We anticipate DiffDiv to be more effective in scenarios where no assumptions about specific shortcuts can be made, where only image data is available, where prompting for diversification is ambiguous, and when working with general (non-canonical) datasets. In the latter, especially, generating samples within the data manifold is often challenging with large pre-trained models. Conversely, when there is a need to mitigate a specific cue and prompts can clearly guide the generation process (e.g., biases related to gender or ethnicity) within standard image distributions, text-to-image models may offer a more targeted approach for addressing that bias. Additionally, our framework benefits from halting DPM training early to facilitate OOD generalization, contrasting with the assumptions typically made by text-to-image models used for diversification.
>
> Finally, it is possible to avoid "silent" failures by monitoring ensemble performance as a proxy for assessing the "OODness" of the generated data. We further elaborate on this important point in our response to (W2 & Q1 (b)).
>
> Following the reviewer's comment we have revised Section 2 to highlight the consideration of alternative generation methodologies and have included all relevant references.
>
> **References**
>
> [1] Howard, P., Madasu, A., Le, T., Moreno, G.L. and Lal, V., 2023. Probing intersectional biases in vision-language models with counterfactual examples. arXiv preprint arXiv:2310.02988.
>
> [2] Venkatraman, S., Jain, M., Scimeca, L., Kim, M., Sendera, M., Hasan, M., Rowe, L., Mittal, S., Lemos, P., Bengio, E. and Adam, A., 2024. Amortizing intractable inference in diffusion models for vision, language, and control. arXiv preprint arXiv:2405.20971.

---

> ### Author Response · Authors · 2024-11-19
> **W1 & Q2 (c) Standard Augmentation**
>
> We have previously explored standard augmentation techniques such as rotations, flips, and random cropping. However, in the specific degenerate datasets we work with, where input features are fully correlated, it is challenging for these methods to yield genuinely new feature combinations (e.g., observing a previously unseen gender-ethnicity pairing). Additionally, these augmentations are commonly used to enhance model performance in classification tasks (we also apply some of these for robust ensemble training), resulting in in-distribution (ID) samples in our context. In DiffDiv, using ID samples for the diversification objective would directly conflict with the classification training objective.

---

> ### Author Response · Authors · 2024-11-19
> **W2 & Q1 (a) Ambiguity of sample classes**
>
> This is a very interesting point! Indeed the class membership is often ambiguous (at least from the perspective of the human eye). This is interesting because we have in fact found that low-fidelity samples aid diversification. We show this in Fig. 4, 6, S3 and S4.
>
> Assume we generate an ambiguous OOD sample which lies in the manifold of the training data. The ensemble of models has pressure to (1) maintain performance on the real data, and (2) assign a label to the ambiguous OOD sample, which has to be diverse across the models in the ensemble.
>
> The intuition behind this work lies within the ability of these ambiguous samples to break the shortcuts (i.e. red is not always associated with a heart, but 'another' shape, even if the shape is ambiguous). We find that for DPMs to be able to achieve this they must be trained enough to capture 'at least' the data manifold, but not so much so as to not be able to generate new combinations.
>
> We have clarified this aspect in the manuscript.

---

> ### Author Response · Authors · 2024-11-19
> **W2 & Q1(b) How to choose the DPM training epoch**
>
> Determining when to stop DPM training to optimize the generative stage is a crucial consideration. Although in our experiments we observed that DiffDiv's performance is not highly sensitive to this choice. For example, we highlight the range of maximum diversity achieved through diversification in Figure 4 and Supplementary Figure S3 (ColorDSprites Fidelity 4-150; UTKFace Fidelity 400-1000; CelebA 50-1200). Notably, strong diversification occurs within approximately one-third of all DPM training steps for most datasets. Generally, we find that DiffDiv converges unless the diffusion model is nearly untrained, or overfit to the distribution (resulting in predominantly in-distribution (ID) samples) (see Figure 6).
>
> Additionally, it is possible to use the ensemble as a proxy to choose the DPM early-stopping regime. We identify the originative stage in the areas achieving the highest change in diversity while keeping the change in accuracy to a minimum. These areas are generally broad and we observe similar DiffDiv performance across all relevant Fidelity levels.
>
> To improve on our original explanations and clarify these important aspects in the paper we have significantly revised and expanded supplementary Section S2.3 and Figure S4. In particular, Figure S4 shows the change in accuracy and diversity by the ensembles trained with DPM samples at different training stages. We further include in the figure the originative search experiments for all datasets. We highlight the areas displaying 'originative' tendencies and the fidelity level chosen for the experiments. The changes have been marked in red in the manuscript.

---

> ### Author Response · Authors · 2024-11-19
> **W3 (a) Data augmentation methods**
>
> We refer the reviewer to the comments W1 & Q2 (b) and (c) for a more detailed answer over the viability of additional augmentations in our framework, and the reason behind our choice of DPMs for counterfactual generation.

---

> ### Author Response · Authors · 2024-11-19
> **W3 (b) Analysis beyond diversification**
>
> Beyond diversification metrics, we show short-cut learning bias mitigation and cue aversion by measuring the fraction of models within the ensemble that attend to primary (shortcut) and non-primary (or non-shortcut) cues. This is achieved by using accuracy as a proxy, and observing which model correctly classifies which cue. We wish to particularly point the reviewer to Table 1, Table 2 and Suppl. Fig. S5 for evidence of a model's aversion to the main shortcut cue for each dataset.
>
> To address the reviewer's concern we extend the original results to further show accuracy-based results on how DiffDiv can lead to bias mitigation and a performance boost over ensemble training.
>
> Firstly, we compute and report the average increase in accuracy for each of the benchmark metrics over the non-shortcut cues, ensuring that the bias mitigation by DiffDiv indeed leads to a more robust feature set for prediction:
>
> | Obj. ↓ Dataset → | Color | Face  | CelebA |
> |--------------------|-------|-------|--------|
> | Cross              | 5.17  | 3.37  | 2.00   |
> | Dis                | 8.16  | 3.61  | 0.45   |
> | KL                 | 2.87  | 4.54  | 2.02   |
> | L1                 | 3.31  | 3.61  | 1.30   |
> | L2                 | 4.67  | 5.04  | 1.34   |
>
>
> We observe an always positive change in accuracy over the averted cues, extending the diversification and cue aversion results in Table 2. We have included these results as a new table in the manuscript.
>
> Secondly, inspired by [1], we have performed ablation studies on model selection post-training. We include in the selected subset any model showing shortcut-cue aversion from Table 2. We also select additional models to reach a dynamic range between 15% and 99% of the original ensemble. We show below the performance of the ensemble after model selection (as opposed to the original performance in parenthesis), and observe a severe performance increase with respect to our baseline results.
> | **Obj. ↓ Dataset →**   | Model Select. Acc. (Ensemble Acc.) <ColorDSprites> | Model Select. Acc. (Ensemble Acc.) <UTKFace> | Model Select. Acc. (Ensemble Acc.) <CelebA> |
> |--------------------------|--------------------------------------------------|--------------------------------------------|-------------------------------------------|
> | Baseline                 | 1.000                                            | 0.943                                      | 0.873                                     |
> | Cross                    | 0.945 (0.856)                                    | 0.856 (0.836)                              | 0.828 (0.745)                             |
> | Dis                      | 0.980 (0.916)                                    | 0.868 (0.826)                              | 0.873 (0.857)                             |
> | KL                       | 0.872 (0.786)                                    | 0.858 (0.837)                              | 0.713 (0.672)                             |
> | Std                      | 0.861 (0.784)                                    | 0.824 (0.816)                              | 0.737 (0.659)                             |
> | Var                      | 0.864 (0.762)                                    | 0.776 (0.757)                              | 0.716 (0.650)                             |
>
> References:
>
> [1] Lee, Y., Yao, H. and Finn, C., 2022. Diversify and disambiguate: Learning from underspecified data. arXiv preprint arXiv:2202.03418.

---

> ### Author Response · Authors · 2024-12-01
>
> We wish to thank again the reviewer for carefully reviewing our work. If any of our responses were satisfactory we would be extremely grateful if you could consider increasing your score. Please feel free to ask additional questions, we would be more than happy to provide further clarifications.

---

### Official Review · Reviewer_EWrp · 2024-11-04

**Soundness:** 2
**Presentation:** 1
**Contribution:** 2
**Rating:** 3
**Confidence:** 4

**Summary:**

This paper tackles the challenge of mitigating shortcut learning by introducing an ensemble framework named DiffDiv, which leverages Diffusion Probabilistic Models (DPMs). DiffDiv generates synthetic counterfactual data through DPMs to weaken shortcut dependencies, enhancing prediction diversity within the ensemble. Experimental results demonstrate that DiffDiv successfully guides the model to perform as intended.

**Strengths:**

- This paper presents a straightforward approach addressing shortcut learning by synthesizing counterfactuals, leveraging the strengths of modern DPMs.

- An in-depth analysis of the impact of sample fidelity on the OODness is nice.

**Weaknesses:**

- While DPMs are central to the proposed method, additional details—such as the training scheme specific to this framework, the model's architecture, and the integration of synthetic data during training—would enhance readability. The current visualization results alone feel insufficient. At least, please provide clear and concise explanations of the overall process for integrating synthetic data into training, ensuring ease of understanding.

- The presentation of the manuscript could be significantly improved. An overview figure of the proposed framework seems essential. Despite the complex components like DPMs in this framework and strategies such as early stopping sampling, the lack of additional explanations makes it challenging for readers to grasp the framework as a whole. Furthermore, the content in Table 1 is difficult to follow. Why are the fractions in Table 1 selected as they are? Also, the reviewer suggests splitting the main quantitative table to highlight the superiority of the proposed method and clearly presenting the ablation study on different diversification objectives for improved readability.

- The paper presents a straightforward solution, but a lack of novelty remains a key concern. While insights like the influence of sample fidelity on OODness and the early-stopping sampling strategy are good, the manuscript would benefit from theoretical analysis if possible, or novel regularizers for DPMs to generate enhanced counterfactuals against spurious correlationsfrom a generative perspective.

- Lastly, the proposed framework requires manual tuning for each specific domain or dataset, making it impractical for real-world application and reproducibility. The manuscript would benefit from introducing a universal cheat sheet or automated tuning strategies that perform effectively across diverse scenarios.

**Questions:**

- The authors consider many objectives for diversificiation such as div, cross, and kl. Is there any reason for this design? If there is an analysis on the selection of diversification objective, it would be better.

- I wonder if the authors have considered the options of using GANs or optimal transport distance to generate synthetic counterfactuals.

---

> ### Author Response · Authors · 2024-11-19
> **Thanks + Q1) Diversification objectives rationale**
>
> Thank you for reviewing our work in depth and providing actionable comments and suggestions. We have made several substantial modifications to improve the clarity and presentation of this work, and address the reviewer's concerns. Please find below our answers to each of your raised points and questions:
>
>
> ### **Q1) Diversification objectives rationale**
>
> We study these objectives for three complementary reasons: first, to benchmark current diversification methodologies in the field, and validate their influence in diversification; second, to show DPM counterfactuals (as explored in DiffDiv) can aid all these methodologies; third, to provide evidence that DiffDiv's workings are in fact valid despite the diversification method used. This is coherent with our central hypothesis, i.e. that samples from appropriately trained DPMs can be used for diversification and bias mitigation. We consider common diversification objectives for model divergence (e.g. kl, L1, and L2), and other objectives from recent literature in ensemble diversification for bias mitigation (e.g. Div [1]). We show ablation experiments in Fig. 5, Fig. 6, Table 1., and Table 2.
>
> Following the reviewer's comment we have expanded the discussion session to discuss our rationale and findings.
>
> **References**
>
> [1] Lee, Y., Yao, H. and Finn, C., 2022. Diversify and disambiguate: Learning from underspecified data. arXiv preprint arXiv:2202.03418.

---

> ### Author Response · Authors · 2024-11-19
> **Q2) GANS or OT as an alternative generative framework**
>
> This is a very interesting question. In DiffDiv we leverage a particular "OODness" phenomenon which to our knowledge is quite unique to Diffusion Probabilistic Models, and which only recently has begun to be studied more in-depth. We do not know of recent work showcasing this phenomena for OT-based models although it would not be surprising if similar results could be achieved.
>
> GANS are an interesting case, although we believe they might be less suited to this task. In GANs the quality of the approximated distribution is shaped by the pressure of the discriminator model to tell apart real and fake samples. Even in adversarial settings, if we assume a degenerate correlated dataset such as in our experiments,  the discriminator classifier may be subject to the same biases as the ensemble models do, which might ultimately influence the final distribution. The concept of "early-stopping" would also no longer apply, and other methods may be necessary to achieve this.
>
> An interesting tangential direction also revolves around distillation methods from the early-stopped DPMs in our experiments. Generating counterfactuals however incurs minimal cost in our experiments, as we generate a small set of 3,000 synthetic samples and validate it to be sufficient to provide effective diversification cues (as supported by ablation experiments in Supplementary Section S2.5 and Table S3).

---

> ### Author Response · Authors · 2024-11-19
> **W1) DPM design and training details.**
>
> Following the reviewer's comment, we have added a new section in the supplementary materials providing details on the denoising architecture used, diffusion training, and counterfactual generation (now Supplementary Methods Section 1.3). We have also revised the methods section to clarify counterfactual generation and integration within  DiffDiv. All changes have been highlighted in red in the manuscript.

---

> ### Author Response · Authors · 2024-11-19
> **W2) Clarity of framework, data integration and improved presentation**
>
> Following the reviewer's comment, we have performed several changes in the manuscript to better detail our framework and improve the readability and clarity of the experiments. Below is a brief summary:
>
> 1. We have separated the experimental framework in Fig.1 into two figures, separately depicting the  DiffDiv framework and the data generation framework.
>
> 2. We have included a new Fig 1. The figure now shows the DiffDiv framework in two phases, DPM training phase, and the ensemble training phase.
>
> 3. We have created a new supplementary figure S1 to only show the experimental framework.
>
> 4. We have added a new Section 2.1 "DiffDiv Overview", to describe the algorithm and tie every subsection in the methods to this description.
>
> 4. We have rearranged all sections in the methods to better explain the algorithm. We start from the DiffDiv-specific sections, i.e. Phase 1: DPM training and Phase 2: Ensemble training, and conclude with the framework used for Data generation and analysis (WCST-ML).
>
> 5. We have revised Table 1 (now Table 2) to improve clarity and readability. As suggested by the reviewer we arranged the ablations on the diversification objective differently and clustered the results achieved with DiffDiv as opposed to classical methods relying on OOD data.
>
> 6. We have significantly extended Supplementary Section S2.2 to clarify the DPM training influence on the generated counterfactuals.
>
> 7. We have revised and extended Figure S4 to show results for all datasets making it clearer how the ensemble validation performance and diversity can be used to select the appropriate DPM training stage for counterfactual generation.
>
> All changes are highlighted in red in the revised manuscript.

---

> ### Author Response · Authors · 2024-11-19
> **W3 (a) DPM regularizers for OOD generation**
>
> The reviewer makes an interesting point and we agree more work is necessary to delve even deeper into the mechanisms of DPMs displaying these interesting characteristics. However, the central narrative of our work revolves around improving current diversification methods, which often rely on expensive data collection techniques.
>
> Rather than devising new regularizers for DPM sample generation, we instead take advantage of an interesting DPM phenomenon, whereby we observe OOD sampling tendencies of DPMs at earlier training stages. We perform extensive experiments to show how we can purposefully induce and take advantage of this phenomenon, and show how it can prove to be a viable substitute for expensive OOD data. Importantly, we ultimately find that appropriate DPM training is sufficient to lead to strong diversification and bias mitigation in ensembles, on par with using real OOD data, reducing the need for additional regularizers or sampling schemes.

---

> ### Author Response · Authors · 2024-11-19
> **W3 (b) Novelty/contributions**
>
> In our work we set forth several key contributions, providing significant advantages over previous methods.
>
> 1. We identify and investigate at length an interesting DPM phenomenon, previously unknown to exist in fully correlated settings.
> 2. We propose a framework leveraging this phenomenon for diversification and shortcut learning mitigation. Importantly this provides a significant advantage over prior work, needing additional data collection to achieve the same results.
> 3. We study in depth the relationship between counterfactual generation, downstream ensemble performance, and shortcut cue aversion. To our knowledge, this has never been studied in depth in this context.
> 4. We show our framework is robust against several diversification objectives, datasets, and experimental settings, providing practical guidelines on how to achieve significant shortcut learning mitigation.
>
> We have also extended our results to include additional performance evaluations, including prediction improvements of the ensemble over the non-shortcut cues, and performance boosts with model selection, mitigating the accuracy diversity trade-offs previously shown. We refer to our comments **(W1)** and **(W2)** to Reviewer **zqYm** for a more detailed overview of the additional results.
>
> Ultimately, our work offers a practical framework for bias mitigation that effectively addresses key limitations of prior methods, while deepening our understanding of DPM-driven diversification, and shortcut learning mitigation. We believe our findings offer meaningful insights and practical benefits for the broader research community. We trust the reviewer will appreciate the significance and potential impact of these contributions.

---

> ### Author Response · Authors · 2024-11-19
> **W4) General guidelines (cheat sheet) to apply DivDiff**
>
> For DiffDis to be successfully applied, we need to generally set two hyperparameters, the diversification $\gamma$, and the DPM fidelity level.
>
> #### **Choice of $\gamma$**
> For the gamma, we report the used values and corresponding curves in Supplementary Figure and Table S1. We find these values to be conditionally dependent mainly on the diversification objective used, and suggest typical settings for each: $\gamma_{L2} = 5.$, $\gamma_{L1} = 2.$, $\gamma_{cross} = 2.$,  $\gamma_{div} = 5.$, and $\gamma_{kl} = 0.2$.
>
> #### **DPM Fidelity**
> Although DPM fidelity is an important factor for good diversification, we observe that DiffDiv is not overly sensitive to the exact choice of DPM fidelity level. For example, we highlight the range of maximum diversity achieved through diversification in Figure 4 and Supplementary Figure S3 (ColorDSprites Fidelity 4-150; UTKFace Fidelity 400-1000; CelebA 50-1200). Notably, strong diversification occurs within approximately one-third of all DPM training steps for most datasets. Generally, we find that DiffDiv converges unless the diffusion model is nearly untrained, or overfit to the distribution (resulting in predominantly in-distribution (ID) samples) (see Figure 6).
>
> Even so, it is indeed possible to automate DPM early stopping. We can identify the originative stage by using the ensemble as a proxy. In our experiments, we find the generative stage to be uniquely denoted by the highest change in ensemble diversity while keeping the negative change in accuracy to a minimum. These areas are generally broad and we observe similar DiffDiv performance across all relevant Fidelity levels.
>
> To address the reviewer's comment and to clarify these important aspects in the paper we have significantly revised and expanded supplementary Section S2.3 and Figure S4. In particular, Figure S4 shows the change in accuracy and diversity by the ensembles trained with DPM samples at different training stages. We further include in the figure the originative search experiments for all datasets. We highlight the areas displaying 'originative' tendencies and the fidelity level chosen for the experiments.

---

> ### Author Response · Authors · 2024-12-01
>
> Thank you again for carefully reviewing our work, and raising pertinent and actionable questions. We hope our answers and changes in the manuscript have addressed your concerns. If any of our responses were satisfactory we would be extremely grateful if you could consider increasing your score. Please feel free to ask additional questions, we would be more than happy to provide further clarifications.

---

### Official Review · Reviewer_zqYm · 2024-11-11

**Soundness:** 3
**Presentation:** 3
**Contribution:** 2
**Rating:** 5
**Confidence:** 3

**Summary:**

The paper addresses the shortcut learning problem. It proposes to use diffusion model to generate new samples which contain new feature combinations. The generated samples will be used to train the ensemble classifiers.

**Strengths:**

The idea of using diffusion models to generate samples with novel feature combination to mitigate short-cut learning makes sense. The paper shows some very interesting findings, such as the relations between the generated sample fidelity vs the diversity. I feel the idea can be useful in mitigating spurious correlations as well.

**Weaknesses:**

1. The paper claims that the proposed method can mitigate short-cut learning. However, the only evidence it provided is the disagreement between ensemble classifier. It’s not very clear how disagreement in ensemble training can lead to mitigating short-cut learning. Similar works (say Lee et al., 2022) leverages worst class acc (or validation accuracy over majority and minor categories) in datasets like Waterbirds and CelebA. These are direct evidence to show how the proposed method can mitigate short-cut learning. Showing experiments like these can greatly strengthen the paper.

2. Similar to above comments, although Table 1 shows that ensemble classifiers trained with DiffDiv generated samples, rely less on the short-cut feature, it also takes a significantly hit on the majority class accuracy. For example, in ColorDSprites dataset, the Acc drops from 100% to 70% - 90% from baseline to the proposed method. It’s hard to justify that the model learns from a more robust feature set. It would be super helpful if the rebuttal can justify why getting a hit on majority class accuracy is not a big concern.

3. The paper is using diffusion model training epochs to evaluate the Fidelity. However, it seems to be a very unstable cue, since 3 different datasets the maximum diversity achieved at very different number of epochs, 20 for ColorDSprites, 800 for UTKFace and 1000 for celebA. In Supp 2.2, the paper mentions that using validation loss can be a good early stopping cue. If it’s the case, why not using validation loss to be a measure? I tried to get some insights from Fig. 6. However, it’s quite hard to understand Fig. 6. Especially 3 datasets showed quite different behavior.

4. From line 242-243, the paper mentions that “L1 and L2 baseline objectives, designed to induce diversity by maximizing the pairwise distance between any two model outputs and the moving average of the ensemble prediction”, however in the loss function, only the average of the prediction is measured. Did I miss anything?

5. It’s not very clear how the diffusion model is trained, especially some important details such at the size of the model and the structure of the model are missing from the paper. It’s also not clear if the conclusion is sensitive to the structure of the diffusion models or not. It would be helpful to clarify the structure of the diffusion model.

**Questions:**

It's not clear to me how disagreement in ensemble training can lead to mitigating short-cut learning. It would be really helpful if the authors can help clarify this in the rebuttal.

I also want to understand the trade-off between the val acc and the disagreement in ensemble training.

---

> ### Author Response · Authors · 2024-11-19
> **Thank you & Q1) Clarifications on how disagreement can lead to shortcut learning mitigation**
>
> We wish to thank the reviewer for the thorough review, the interest in the work, and the insightful comments. We have revised our manuscript following the reviewer's questions and suggestions. Please find below our answer to each of your points:
>
>
> ### **Q1) Clarifications on how disagreement can lead to shortcut learning mitigation**
>
> Consider an ensemble of models trained on a labeled predictive task. Suppose that in the training data, both the color and shape of objects within each image could perfectly explain the mapping between inputs and outputs. Typically, the ensemble would learn to rely on the simplest cue—color—across all models.
>
> Now, let’s assume we have access to additional, unlabeled (OOD) data that presents previously unseen combinations of colors and shapes. Without further intervention, each model in the ensemble would tend to assign a label to these new data points based on color, although we don’t know if this choice is correct according to the training data. Introducing a diversification objective on these OOD samples changes this dynamic: if the first model in the ensemble assigns a label based on color, the **diversification objective** will encourage the second model to consider an alternative label, perhaps one more aligned with shape (which must remain consistent with the training data by the **classification objective**). This diversity across models is key for reducing shortcut reliance.
>
> It’s crucial that the samples used for the diversification objective present novel feature combinations. If they were in-distribution (ID) samples, the diversification objective would directly conflict with the classification objective on the original training data, undermining the effect.
>
> In our approach, rather than assuming access to such data, we leverage a unique property of DPMs to generate these OOD samples, showing when and how they effectively foster strong diversification in the ensemble.
>
> We have revised Section 2.1 to clarify this aspect in the manuscript.

---

> ### Author Response · Authors · 2024-11-19
> **Q2) Trade off between val acc and the disagreement in ensemble training**
>
> In principle, each model is trained to accurately predict the original data while also diversifying on additional OOD data. While this approach generally ensures diversity and encourages many models to shift their focus toward more robust, non-primary cues, it does not guarantee that all models will follow suit. Some may still end up relying on other spurious cues, which can negatively impact performance. In our response to point (W2) below, we delve further into this phenomenon and demonstrate how it can be mitigated through model selection, along with extended results to support this claim.
>
> To provide greater clarity, we have revised the methods and results sections to better illustrate the interplay between diversification and the trade-off with validation accuracy. All relevant changes have been marked in red in the revised manuscript.

---

> ### Author Response · Authors · 2024-11-19
> **W1) Short-cut learning mitigation beyond disagreement**
>
> This is a crucial point. Beyond diversification metrics, we demonstrate the mitigation of shortcut learning biases and cue aversion by measuring the proportion of ensemble models that focus on primary (shortcut) versus non-primary (non-shortcut) cues. We use accuracy as a proxy, evaluating which models correctly classify each cue. We direct the reviewer’s attention to Table 2 and Supplementary Figure S5, which illustrate the model's aversion to the primary shortcut cue across datasets.
>
> Furthermore, inspired by the reviewer's comments, we analyzed the extent to which the ensemble's performance improved over non-shortcut cues. We computed the average accuracy of the ensemble across averted cues when trained with classical methods versus using DiffDiv. The results, presented below, show the average change in accuracy across various metrics with DiffDiv:
>
> | Obj. ↓ Dataset → | Color | Face  | CelebA |
> |--------------------|-------|-------|--------|
> | Cross              | 5.17  | 3.37  | 2.00   |
> | Div                | 8.16  | 3.61  | 0.45   |
> | KL                 | 2.87  | 4.54  | 2.02   |
> | L1                 | 3.31  | 3.61  | 1.30   |
> | L2                 | 4.67  | 5.04  | 1.34   |
>
> Importantly, we consistently observe a positive change in accuracy over the averted cues, indicating increased reliance on these cues for accurate prediction and further strengthening the diversification and cue aversion results shown in Table 2. These findings have been incorporated as a new table in the manuscript (Table 1).

---

> ### Author Response · Authors · 2024-11-19
> **W2)  Negative inpact on majority class**
>
> Mitigating shortcut learning via ensemble diversification often entails a performance drop on majority classes, this is a finding that is coherent with existing literature. A prominent methodology to mitigate this phenomenon is ensemble model selection, where a subset of models is selected for final ensemble inference. To address the reviewer's concern we have performed further tests to show the impact of model selection to mitigate this phenomenon in our framework. To ensure diversity in the final selection, we include in the selected subset any model showing shortcut-cue aversion from Table 2. Furthermore, we select additional models to reach a dynamic range between 15% and 99% of the original ensemble. We show below the performance of the ensemble after model selection (as opposed to the original performance in parenthesis).
>
> | **Obj.↓**   Dataset→     |  Model Select. Acc. (Ensemble Acc.) <ColorDSprites> |  Model Select. Acc. (Ensemble Acc.) <UTKFace> | Model Select. Acc. (Ensemble Acc.) <CelebA>|
> |-----------------|-----------------------------------|-----------------------------------|-----------------------------------|
> | Baseline        | 1.000                      | 0.943                     | 0.873                      |
> | Cross           | 0.945 (0.856)                     | 0.856 (0.836)                     | 0.828 (0.745)                     |
> | Dis             | 0.980 (0.916)                     | 0.868 (0.826)                     | 0.873 (0.857)                     |
> | KL              | 0.872 (0.786)                     | 0.858 (0.837)                     | 0.713 (0.672)                     |
> | Std             | 0.861 (0.784)                     | 0.824 (0.816)                     | 0.737 (0.659)                     |
> | Var             | 0.864 (0.762)                     | 0.776 (0.757)                     | 0.716 (0.650)                     |
>
> We observe a significant increase in performance, severely mitigating this phenomena, while retaining model diversity.
>
> We have updated the manuscript following the reviewer's comment. We have added a new supplementary Section (S2.2) and a version of the table above (Table S2) to the supplementary materials to address this point in the manuscript.

---

> ### Author Response · Authors · 2024-11-19
> **W3) A cue for diffusion 'early stopping'**
>
> Determining when to stop DPM training to optimize the generative stage is a crucial consideration. However, in our experiments, we observed that DiffDiv's performance is not highly sensitive to this choice. For example, we highlight the range of maximum diversity achieved through diversification in Figure 4 and Supplementary Figure S3 (ColorDSprites Fidelity 4-150; UTKFace Fidelity 400-1000; CelebA 50-1200). Notably, strong diversification occurs within approximately one-third of all DPM training steps for most datasets. Generally, we find that DiffDiv converges unless the diffusion model is nearly untrained, or overfit to the distribution (resulting in predominantly in-distribution (ID) samples) (see Figure 6).
>
> We selected "fidelity" for qualitative and analytical exploration to support our hypothesis, and show how "early-stopping" can be used effectively leverage this phenomenon. We acknowledge that, algorithmically, this may not be the most intuitive depiction, as noted by the reviewer.
>
> To improve on our original explanations and clarify these important aspects in the paper we have significantly revised and expanded supplementary Section S2.3 and Figure S4. In particular, Figure S4 shows the change in accuracy and diversity by the ensembles trained with DPM samples at different training stages. We further include in the figure the originative search experiments for all datasets. We highlight the areas displaying 'originative' tendencies and the fidelity level chosen for the experiments.  We identify the originative stage in the areas achieving the highest change in diversity while keeping the change in accuracy to a minimum. These areas are generally broad and we observe similar DiffDiv performance across all relevant Fidelity levels. In Figure 6, these correspond to the settings achieving metrics closest to the top right corner in the figure. We have also revised several paragraphs in the manuscript to reflect and better highlight this point. The changes have been marked in red in the manuscript.

---

> ### Author Response · Authors · 2024-11-19
> **W4) L1 and L2 objectives**
>
> Thank you for catching this; indeed, it was a misstatement in the text. The L1 and L2 losses specifically maximize the standard deviation and variance of the ensemble models' predictions, respectively. Pairwise distances are considered within other metrics. We have revised the relevant section to provide a clearer explanation of this aspect.

---

> ### Author Response · Authors · 2024-11-19
> **W5) Diffusion model training details**
>
> We have added a section in the supplementary materials providing further details on the denoising architecture used, diffusion training, and counterfactual generation (now Suppl. Methods Section $\S$ S1.3).

---

> ### Author Response · Authors · 2024-11-27
>
> Thank you once again for your thorough evaluation. We hope our answers and changes in the manuscript have addressed your concerns. We are more than happy to provide further clarifications if needed.

---

> > ### Comment · Reviewer_zqYm · 2024-11-27
> > **Thank you**
> >
> > Thanks a lot for your reply and clarification. My questions has been answered well. I will consider the new changes in the final score. Thank you again for the detailed reply.

---

### Author Response · Authors · 2024-11-21
**Response to all Reviewers**

We sincerely thank all reviewers for their constructive feedback and thorough evaluation of our work. We are particularly grateful for the recognition of our approach as **novel** (**GGVZ**) and the application of diffusion models to mitigate shortcut learning as **promising** (**Fptv**), **practical** (**GGVZ**), **useful** (**zqYm**) and with **significant implications** for real-world scenarios (**GGVZ**). The reviewers appreciated the **innovative use** of DPM-generated counterfactuals to promote ensemble diversification (**GGVZ**) and the analysis of diversity and shortcut cue aversion **in depth** (**EWrp**), **thorough and insightful** (**Fptv**).

We have made several substantial changes to the manuscript to address the reviewers' comments and suggestions. Detailed responses to each comment are provided in our replies to the individual reviewers. Below, we provide a concise summary of the main revisions made:

### **Summary of Changes**

1. **Clarified Ensemble Disagreement and Shortcut Mitigation:**
   - Expanded the Methods sections to explain how ensemble disagreement reduces shortcut reliance by leveraging novel feature combinations.

2. **Extended Experimental Results (new results):**
   - Added Table 1 showcasing accuracy improvements over non-shortcut cues to demonstrate robust feature learning.
   - Added Table S2 showcasing model selection experiments post-training to mitigate accuracy trade-offs.

3. **Improved Framework Clarity:**
   - Introduced Section 2.1 ("DiffDiv Overview") for a structured framework description.
   - Reorganized Table 2.
   - Reorganized the Methods section for clarity and flow.
   - Split Figure 1 into two figures to distinguish DiffDiv (Figure 1) and data generation processes (Figure S1).

4. **Enhanced Reproducibility and Methodology:**
   - Added a new supplementary section (S1.3) detailing DPM architecture, training, and counterfactual generation steps.

5. **Addressed DPM Training Sensitivity and Early Stopping:**
   - Expanded Section S2.3 and Figure S4 to illustrate effective DPM training ranges and how validation performance informs early stopping.

6. **Broader Context,  Comparisons and others:**
   - Discussed the strengths and limitations of alternative methods in relation to DiffDiv.
   - Added references to relevant concurrent work for context.
   - Fixed minor text inconsistencies.


We believe these updates address all concerns and significantly improve the manuscript. Thank you again for your thoughtful reviews.

---

### Meta-Review · Area_Chair_s7Jo · 2024-12-21

**Metareview:**

The paper addresses an important challenge - mitigating shortcut learning - by exploiting the capacity of DPMs to generate feature-compositional counterfactuals.

Reviewers like the originality of the idea, the clarity of writing, and the potential for practical application in various domains.  However, reviewers remain concerned about 1) the lack of direct, standard evaluations for shortcut learning, 2) the computational expense and tuning challenges (e.g., best epoch for diffusion), 3) the explanation of how ensemble disagreement measurably leads to less reliance on shortcut cues, 4) missing methodological details (DPM architecture, hyperparameter selection) and more direct comparisons to alternative augmentation or counterfactual-generation approaches.

It received 4 divergent reviews, with ratings 5,5,6,3.  Reviewers acknowledged that the rebuttal fixed many details in the original submission and significantly improved the quality.  However, they were only convinced that the proposed method induces diversity, but not that it actually mitigates shortcut learning, since critical side-by-side comparisons on the worst-category accuracy against SOTA were missing.  Therefore, the final ratings remain the same, and overall reviewers agree on the major concern and consider the paper below the bar for publication.

**Additional Comments On Reviewer Discussion:**

While there is no change in ratings post-rebuttal, reviewers acknowledged that the rebuttal fixed many confusing details in the original submission and significantly improved the quality.  Their main concern is that the paper only demonstrated diversity through design but not effectiveness on mitigating short-cut learning, due to missing performance comparisons over SOTA on the worst-category accuracy.

---

### Decision · Program_Chairs · 2025-01-22

Reject